# Implicit Task-Driven Probability Discrepancy Measure for Unsupervised Domain Adaptation

**Mao Li,      Kaiqi Jiang,      Xinhua Zhang**
Department of Computer Science, University of Illinois at Chicago
Chicago, IL 60607
{mli206,kjiang10,zhangx}@uic.edu

## Abstract

Probability discrepancy measure is a fundamental construct for numerous machine learning models such as weakly supervised learning and generative modeling. However, most measures overlook the fact that the distributions are not the end-product of learning, but are the input of a downstream predictor. Therefore, it is important to warp the probability discrepancy measure towards the end tasks, and towards this goal, we propose a new bi-level optimization based approach so that the two distributions are compared not uniformly against the entire hypothesis space, but only with respect to the optimal predictor for the downstream end task. When applied to margin disparity discrepancy and contrastive domain discrepancy, our method significantly improves the performance in unsupervised domain adaptation, and enjoys a much more principled training process.

## 1   Introduction

Discrepancy measures on two distributions underpin a large variety of machine learning tasks, and have been studied extensively since the dawn of modern probability [1]. For example, in generative models, such a measure is applied to align the generated distribution with the empirical one, and prevalent examples include 1) the $f$-divergence that admits a convenient variational form hence can be effectively evaluated via sample-based adversarial optimization [2, 3]; 2) integral probability metric [IPM, 4] that seeks the largest discrepancy in function expectation over a reproducing kernel Hilbert space (RKHS) [MMD GAN, 5–7], 1-Lipschitz continuous functions [Wasserstein GAN, 8, 9], or unit $L_2$ norm functions [Fisher GAN, 10], etc.

In domain adaptation [DA, 11, 12], probability discrepancy is also the key construct in the feature adaptation approach, where a feature extractor $\phi$ is sought to align the source and target distributions transformed by $\phi$ [13, 14]. The aforementioned measures can be applied directly in this context.

It has been long noted that the discrepancy should be tailored to the function class of interest, e.g., those for which we would like to compute expectations. This principle has been applied to density estimation [15] amongst others, where the RKHS is selected to match the downstream task such as image categorization based on the compressed pixel distribution. Naturally this motivation can be easily implemented in IPMs by customizing the generating function space.

However such tailoring remains oblivious to the loss and available labels of the end task. Intuitively, if the latent features in DA are to be used for classification, then whether the loss is AUC or F-score should ideally influence the probability discrepancy. The seminal $\mathcal{H}\Delta\mathcal{H}$-divergence is designed for classification accuracy [16], with a few extensions to Bayesian and other losses [17–20]. Despite being data-dependent, however, they are *unsupervised* without accounting for the available labels. Likewise, if a generative model is used to augment data so as to improve segmentation accuracy [21], then the adversarial network in GANs should not only be able to distinguish between real and synthetic, but also "align", in an appropriate sense, with the segmentation labels at hand.

35th Conference on Neural Information Processing Systems (NeurIPS 2021).

Warping probability discrepancies towards a task has been lightly touched in unsupervised DA (UDA). [22] trains two classifiers that not only boost the source-domain accuracy, but also maximally disagree on the target domain. Unfortunately, it is only formulated as a procedure, *not* a probability discrepancy. Most relevant to our work is the margin disparity discrepancy [MDD, 23], which is based on the $\mathcal{H}\Delta\mathcal{H}$-divergence where two fictitious classifiers $h$ and $h'$ are jointly optimized to maximally reveal the two distributions' difference. [23] took the key insight that $h$ can be tied with the source-domain predictor, and can thus be optimized to simultaneously reduce the source-domain risk and the $\mathcal{H}\Delta\mathcal{H}$-divergence. However, in spite of its effectiveness in both theory and practice, we discover that the specific formulation conflicts with the $\mathcal{H}\Delta\mathcal{H}$-divergence — the latter tries to *maximize* over $h$ so as to promote the divergence, while MDD tries to *minimize* it (Section 3). This undermines the power of MDD in distinguishing two distributions as illustrated in Figure 1. Flipping the sign and min/max cannot resolve the issue.

Our first contribution, therefore, is to develop a new task-driven discrepancy framework that overcomes this obstacle. The key inspiration is that MDD relies on the *pseudo-label* in the target domain (i.e., speculation of their labels based, e.g., on the source-domain head), and this is also the case for some other measures such as the contrastive domain discrepancy [CDD, 24], which promotes the proximity between the class mean of the two domains for each class, and pushes apart the mean of different classes. Such a commonality motivates us to generate the target-domain pseudo-label based on the *optimal* source-domain classifier $h^*$. In MDD, this provides a natural substitute for the fictitious classifier $h$ (Section 3.2) which *no longer needs to be optimized over*, thereby solving the aforementioned problem. As our second contribution, we extend this strategy to CDD in Section 4. The overall formulation becomes a bi-level optimization solvable by implicit differentiation (hence the modifier "implicit" in the method's name).

We note in passing that pseudo-label is commonly used in self-training for UDA [25–28]. However, most methods require various refinements of it in order to mitigate its inaccuracy due to distributional shift [29]. Examples include label sharpening [30], entropy reweighting [31], cycle training [29]. We instead directly use the output of $f^*$ as the pseudo-label in probability discrepancy, outperforming state of the art on a range of datasets (Section 6).

UDA has recently received considerable interest, and most algorithms rely on ad-hoc heuristics; we will mention a few below. Many of them require perusing the code and configuration script. As such, our main goal is *not* to develop yet another highly engineered model that performs better, but to present a *principled* formulation solvable by *off-the-shelf* optimizers. Although our implicit task-driven discrepancy can be straightforwardly applied to generative models, we deem it a better use of space to fully demonstrate its power in UDA. Such a probability discrepancy can also be easily extended to measure (conditional) independence, which has witnessed immediate application in fair and disentangled representation learning [32–35].

## 2 Preliminaries

In UDA, there is a source domain and a target domain, and they are respectively represented as a joint distribution $S$ and $T$ on an input-output space $\mathcal{X} \times \mathcal{Y}$. We will denote their marginal distributions via subscripts, e.g., $S_x$ and $T_y$. The $\mathcal{Y}$ domain can be multiclass with labels $[C] := \{1, 2, \ldots, C\}$. We are provided with labeled examples in the source domain, denoted as an empirical distribution $\tilde{S}$. On the target domain, however, we can only access unlabeled examples, i.e., an empirical distribution $\tilde{T}_x$ which only encompasses the input part of an empirical distribution $\tilde{T}$. In short, let the empirical distributions consist of $\{x_i^s, y_i^s\}_{i=1}^{n_s}$ and $\{x_j^t\}_{j=1}^{n_t}$ for the source and target domains respectively.

The goal of UDA is to find a classifier that predicts well on the target domain $T$. This is often referred to as inductive learning, while, in contrast, transductive learning is only concerned with the prediction on the empirical distribution $\tilde{T}$, whose feature component $\tilde{T}_x$ is available at training time.

The classification model, shared by both domains, consists of a feature extractor (e.g., ResNet) parameterized by $\phi$ and a head $h_\theta$ parameterized by $\theta$. Letting $\ell$ be the loss over the ground-truth label $y$ and the prediction $h_\theta(\phi(x))$, we seek the $\phi$ and $\theta$ that minimize the target-domain risk

$$\mathbb{E}_{(x,y)\sim T}\, \ell(y, h_\theta(\phi(x))), \quad \text{or its empirical counterpart} \quad \mathbb{E}_{(x,y)\sim \tilde{T}}\, \ell(y, h_\theta(\phi(x))). \quad (1)$$

In order to leverage the labeled data from the source domain and the unlabeled target-domain data, the feature adaption approach enforces low empirical risk on the source domain (thanks to the availability

of labels there) and encourages that the source domain distribution, after being transformed by the feature extractor $\phi$, "aligns" well with that of the target domain [13, 14, 36, 37]. This is achieved by

$$\min_{\phi,\theta} \mathbb{E}_{(x,y)\sim\tilde{S}} \ell(y, h_\theta(\phi(x))) + d(\phi\#\tilde{S}_x, \phi\#\tilde{T}_x), \tag{2}$$

where $\phi\#\tilde{S}_x$ is the pushforward distribution of $\tilde{S}_x$, and $d$ denotes some discrepancy measure between two distributions. The intuition is that by "mixing" the latent distributions across the two domains through $\phi$, the favorable accuracy of $h_\theta$ on the source domain can be transferred to the target domain. For simplicity, we will denote $P := \phi\#S_x$ and $\tilde{P} := \phi\#\tilde{S}_x$, and explicitize its dependency on $\phi$ by writing $P_\phi$ whenever necessary. With $z = \phi(x)$, we can derive a conditional distribution of $y$ given $z$ based on $S$, and we denote it as $S_{y|z}$. Similarly, let $Q := \phi\#T_x$, $\tilde{Q} := \phi\#\tilde{T}_x$, and define $T_{y|z}$ analogously.

## 3 Implicit Task-Driven Margin Disparity Discrepancy

There has been a plethora of research on sample-based discrepancy measure between two distributions. Examples include maximum mean discrepancy [MMD, 38], and (neural) variational optimization [39] which effectively subsumes a number of adversarial learning based measures [2, 14].

However, these methods are oblivious to the subsequent tasks that are based on $P$ and $Q$. For example, UDA can be aimed to classify well on these distributions. In domain-adversarial neural networks [DANN, 14], the discrepancy between $P$ and $Q$ is measured via the Jensen-Shannon divergence, reformulated in an adversarial objective as in the generative adversarial network [GAN, 40]. Moreover, MMD simply measures the largest possible difference in the function expectation over $P$ and $Q$:

$$\mathrm{MMD}(P,Q) := \sup_{f\in\mathcal{H}:\|f\|_{\mathcal{H}}\leq 1} \left[ \mathbb{E}_{x\sim P} f(x) - \mathbb{E}_{x\sim Q} f(x) \right] = \left\| \mathbb{E}_{x\sim P} k(x,\cdot) - \mathbb{E}_{x\sim Q} k(x,\cdot) \right\|_{\mathcal{H}}, \tag{3}$$

where $\mathcal{H}$ is the reproducing kernel Hilbert space (RKHS) induced by a kernel $k$. Obviously, it does not take into account whether $f$ is used for classification or regression. The celebrated $\mathcal{H}\Delta\mathcal{H}$-divergence addresses this problem by focusing on binary classification [16, Lemma 3]:

$$d_{\mathcal{H}\Delta\mathcal{H}}(P,Q) := \max_{h\in\mathcal{H}} \max_{h'\in\mathcal{H}} \mathcal{D}(h, h', P, Q), \tag{4}$$

$$\text{where} \quad \mathcal{D}(h, h', P, Q) := |\mathbb{E}_P[\![\mathrm{sign}\circ h' \neq \mathrm{sign}\circ h]\!] - \mathbb{E}_Q[\![\mathrm{sign}\circ h' \neq \mathrm{sign}\circ h]\!]|. \tag{5}$$

Here $\mathrm{sign}\circ h$ applies the sign function on the output of $h$. $\mathcal{H}$ is a hypothesis space (not necessarily an RKHS), and $[\![\cdot]\!]$ is the Iverson bracket that evaluates to $1$ if $\cdot$ is true, and $0$ otherwise. However, it still does not concern the label of the data (i.e., align only in an unsupervised fashion). To warp the measure to the end-task in a data-dependent fashion, [23] proposed the margin disparity discrepancy (MDD), which improved upon [22] by formulating a principled objective function instead of a heuristic procedure. According to Equation 24 of [23], MDD employs

$$d_{\mathrm{MDD}}(P,Q) = \min_{h\in\mathcal{H}} \left\{ \mathcal{R}(h; P) + \max_{h'\in\mathcal{H}} \mathcal{D}(h, h', P, Q) \right\}, \tag{6}$$

$$\text{where} \quad \mathcal{R}(h; P) := \mathbb{E}_{z\sim P, y|z\sim S_{y|z}} \ell(h(z), y) + \mathrm{reg}(h) \quad \text{is the regularized risk,} \tag{7}$$

and the 0-1 loss in $\mathcal{D}$ can be replaced by smooth surrogates such as hinge loss or cross-entropy loss. Here reg is any standard regularizer applied in regularized risk minimization, e.g., $\ell_2$ norm. The underlying insight is that when comparing $P$ and $Q$, one only needs to consider those $h$ that predict well on the (labeled) source domain, while leaving $h'$ to reveal the maximum discrepancy between $P$ and $Q$. Similar ideas have been leveraged in [22, 41].

### 3.1 Conflict between MDD and $\mathcal{H}\Delta\mathcal{H}$-divergence

Unfortunately, $d_{\mathrm{MDD}}$ turns out conflicting with the spirit of $\mathcal{H}\Delta\mathcal{H}$-divergence in an important way. Note that $h$ is *maximized* in $\mathcal{D}$ as in (4), while it is *minimized* in $d_{\mathrm{MDD}}$ as in (6). This raises a natural question: can the distribution discrepancy be sufficiently revealed when $\max_h$ is replaced by $\min_h$ in the definition of $\mathcal{D}$, i.e.,

$$d_{\mathcal{H}\Delta\mathcal{H}}^{\min}(P,Q) := \min_{h\in\mathcal{H}} \max_{h'\in\mathcal{H}} \mathcal{D}(h, h', P, Q). \tag{8}$$

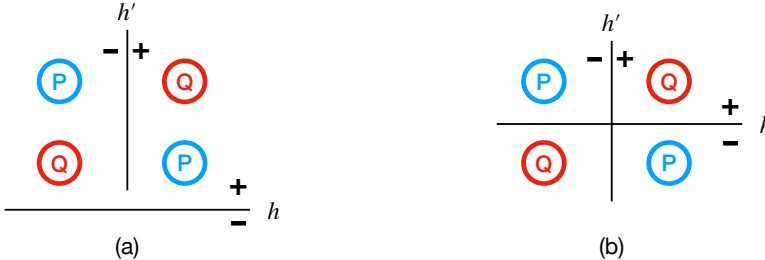

(a)                                           (b)

Figure 1: An example showing that changing $\max_h$ into $\min_h$ undercuts the power of discriminating two distributions. Here the source distribution $P$ has two blue clusters, and the target distribution $Q$ consists of two red clusters. The location of $h$ in (a) makes $\max_{h' \in \mathcal{H}} \mathcal{D}(h, h', P, Q) = 0$, meaning that the new discrepancy $d^{\min}_{\mathcal{H}\Delta\mathcal{H}}(P, Q)$ cannot distinguish the two distributions. In contrast, the $h$ in (b) makes $\max_{h' \in \mathcal{H}} \mathcal{D}(h, h', P, Q) = 1$, implying that the original $d_{\mathcal{H}\Delta\mathcal{H}}(P, Q)$ can distinguish.

It turns out such a change does undermine the discriminative power, and an example is illustrated in Figure 1. Here both the source and target domains have two separate clusters, and the hypothesis space is the horizontal or vertical half spaces (i.e., decision stumps). Sub-figure (a) shows that the minimum $h$ in $d^{\min}_{\mathcal{H}\Delta\mathcal{H}}$ is attained at the horizontal line, and it is easy to check that no matter where $h'$ is placed, $\mathcal{D}(h, h', P, Q) = 0$. In contrast, the $h$ and $h'$ shown in (b) attain $\mathcal{D}(h, h', P, Q) = 1$. So changing maximization of $h$ into minimization caused significant loss in the discrimination power. A more detailed discussion in conjunction with $\mathcal{R}$ as in (6) is available in Appendix A.

### 3.2  A new implicit task-driven MDD

Flipping back the optimization of $h$ turns out far more involved that it appears. It cannot be achieved by simply changing $\min_h$ into $\max_h$ in (6) with the source domain risk negated:

$$\max_{h \in \mathcal{H}} \left\{ -\mathcal{R}(h; P) + \max_{h' \in \mathcal{H}} \mathcal{D}(h, h', P, Q) \right\}, \tag{9}$$

This is because $P$ and $Q$ indeed depend on the feature extractor $\phi$. If we next minimize $d_{\text{MDD}}(P, Q)$ over $\phi$, then it implicitly promotes the source domain risk. If $d_{\text{MDD}}(P, Q)$ is instead maximized over $\phi$, then $\phi$ would attempt to increase the discrepancy $\mathcal{D}$. With a few trials, it becomes clear that the same issue persists in other combinations of flipping sign or min/max.

Our first contribution, hence, is to resolve this issue by turning $d_{\text{MDD}}$ into a *constrained* formulation:

$$\max_{h \in \mathcal{H}: \mathcal{R}(h; P) \leq \lambda} \max_{h' \in \mathcal{H}} \mathcal{D}(h, h', P, Q), \tag{10}$$

where $\lambda$ is some pre-specified cap of loss. Constraining the performance of a classifier is quite commonly used in, e.g., gradient episodic memory to combat catastrophic forgetting [GEM, 42, 43]. However, GEM only solves a linear approximation instead of the exact problem, and it is arguably difficult to *differentiate through* for optimizing $\phi$ in (10). Therefore, we finally develop a bi-level optimization by setting $h$ to the optimal one for the source domain, and then using it in the discrepancy measure. We call it $i$-MDD because it will rely on implicit differentiation for training. The overall training objective can be written as:[1]

$$\min_{\phi} d_{i\text{-MDD}}(\tilde{P}_{\phi}, \tilde{Q}_{\phi}) + \alpha \mathcal{R}(h^*; \tilde{P}_{\phi}) \quad \text{where} \quad d_{i\text{-MDD}}(\tilde{P}_{\phi}, \tilde{Q}_{\phi}) := \max_{h' \in \mathcal{H}} \mathcal{D}(h^*, h', \tilde{P}_{\phi}, \tilde{Q}_{\phi}), \tag{11}$$

$$h^* := \arg\min_{h \in \mathcal{H}} \mathcal{R}(h; \tilde{P}_{\phi}). \tag{12}$$

Here $\alpha > 0$ is a tradeoff parameter. If we do not include $\mathcal{R}(h; \tilde{P}_{\phi})$ in the objective, then the feature $\phi$ would receive no incentive to reduce the source-domain risk. This term in the objective function does not necessitate new implicit differentiation, because $h^*$ is exactly the minimizer of $\mathcal{R}(h; \tilde{P}_{\phi})$. The architecture of $i$-MDD is shown in Figure 2, in comparison with MDD.

---

[1]One might wonder why the max over $h$ in (10) is turned into min over $h$ in (12). This is because $\lambda$ is set to $\min_h \mathcal{R}(h; P)$. Analogously, maximizing $f(x, y)$ over $(x-1)^2 \leq 0$ is equivalent to evaluating $f(1, y)$, because 0 is the minimum of $(x-1)^2$ attained at $x = 1$. Or to bear more resemblance to (11) and (12), it is equal to $f(x^*, y)$ where $x^* = \text{argmin}_x (x-1)^2$.

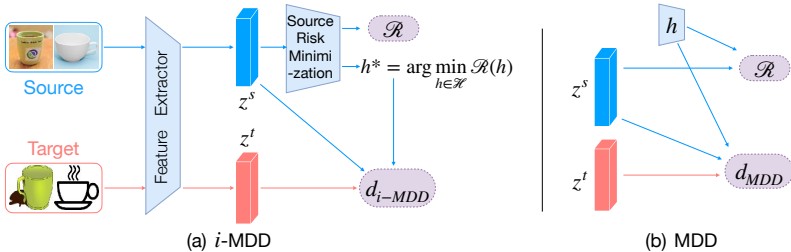

Figure 2: Illustration of $i$-MDD and MDD. The $h^*$, fed into $d_{i\text{-MDD}}$ in $i$-MDD, is the minimizer of $\mathcal{R}$.

### 3.3 Practical discussions: differentiable surrogates

Since the 0-1 loss in $\mathcal{D}$ is not amenable to differentiable training, we follow [23] to morph it into the cross-entropy loss (CE). In particular, suppose $h$ outputs a $C$ dimensional logit vector, and $p = \text{softmax}(h)$. Similarly, $p' = \text{softmax}(h')$. Then the standard $\text{CE}(p', p) = -\sum_i p_i \log p_i' \geq 0$. To combat exploding or vanishing gradient, [3] proposed a modified CE: $\text{MCE}(p', p) = \sum_i p_i \log(1 - p_i') \leq 0$. Then [23] adopts the approximation

$$\mathcal{D}(h, h', \tilde{P}_\phi, \tilde{Q}_\phi) \approx \mathbb{E}_{\tilde{Q}_\phi} \text{MCE}(p', ind \circ p) - \gamma \mathbb{E}_{\tilde{P}_\phi} \text{CE}(p', ind \circ p), \quad (\gamma > 0) \tag{13}$$

where $ind : \mathbb{R}^C \to \{0, 1\}^C$ is the indicator function mapping a vector $v$ to the $i^*$-th canonical vector with $i^* = \arg\max_i v_i$. In practice, the formulation has two issues. First, the right-hand side of (13) is unbounded from below, making it possible for $\phi$ (the minimizing variable) to push it to the negative infinity when solved by stochastic saddle-point optimization. As a result, the implementation of [23] tuned the step size delicately. Secondly, the indicator function $ind$ blocks the backpropagation through the branch of $h$, jeopardizing the proper optimization. We tried removing the indicator function but observed negative infinity even after finely tuning the step size.

In contrast, our new $i$-MDD is immune to these issues, where (13) is used without including the indicator function. In our experiment, we observed that the head $h_\theta$ only needs to be linear in order to achieve state-of-the-art performance. This provided considerable convenience because the optimization for $h^*$ in (12) can be accomplished very efficiently with high precision by convex solvers such as LIBLINEAR [44]. Similarly, it is clear that $h^*$ does *not* depend on $h'$, but on $\tilde{P}_\phi$ only (i.e., $\phi$). Therefore, we can first solve $h^*$ in (12), and then fix it when solving $h'$ in (11), which results in another convex problem thanks to the linearity of $h'$. These conveniences significantly benefit computation and convergence properties.

Although MDD can forgo the stochastic saddle-point optimization and also evaluate $d_{\text{MDD}}$ exactly, the inner *joint* maximization over $h$ and $h'$ leads to a non-concave function, hence impairing the precision of backpropagation. Even if the indicator function is imposed and optimization is only over $h'$, we found a linear $h'$ was insufficient to deliver accurate predictions.

### 3.4 Bi-level optimization

Bi-level optimization has recently received intensive study [45–48], and they can be easily applied to $i$-MDD. Thanks to the linearity of $h$ and $h'$, the backpropagation can be performed in a *closed form*. Denote the ultimate objective value in (11) as $J$. Letting $z_i^s = \phi(x_i^s)$ and $z_j^t = \phi(x_j^t)$, we only need to derive new strategies to compute $\partial J/\partial z_i^s$ and $\partial J/\partial z_j^t$, based on which backpropagation through the feature extractor will be standard. Towards this end, most of the implicit differentiation approaches rely on multiplying a given vector to the Hessian of the loss $\ell$ in (12) with respect to $h$ [45]. Interestingly, for linear multi-class classifiers with cross-entropy loss, the formula has already been derived by [49, Appendix D], and we quote their results in Appendix B for completeness, along with the detailed analysis of computational complexity.

To summarize, the crux of $i$-MDD is to replace the $h$ in the $\mathcal{H}\Delta\mathcal{H}$-divergence by the optimal source domain classifier $h^*$ under the current $\phi$. This is in line with the pseudo-label approach and $h^*$ can be applied to the target domain to provide a soft label. Indeed this principle can be applied to other class-aware discrepancy measures, and our next contribution is to warp the contrastive domain discrepancy [CDD, 24] towards the end task.

# 4 Task-driven Contrastive Domain Discrepancy

Underpinning CDD is the hard pseudo-label $\hat{y}_j^t \in [C]$ assigned to each target domain example $z_j^t$. [24] adopted clustering on $z_j^t$, where each class corresponds to a cluster, and its center is initialized by the mean of the source domain $z_i^s$. Naturally, $\hat{y}_j^t$ is set to the cluster that $z_j^t$ belongs to at convergence. Then the discrepancy between $\tilde{P}$ and $\tilde{Q}$ is defined as (distilled from Equations 3 and 4 in [24])

$$d_{\text{CDD}}(\tilde{P}, \tilde{Q}) = \underbrace{\frac{1}{C} \sum_{c \in [C]} \left\| \mu_c^s - \mu_c^t \right\|_{\mathcal{H}}^2}_{\text{intra-class discrepancy}} - \beta \cdot \underbrace{\frac{1}{C(C-1)} \sum_{c \neq c'} \left\| \mu_c^s - \mu_{c'}^t \right\|_{\mathcal{H}}^2}_{\text{inter-class discrepancy}}, \tag{14}$$

$$\text{where} \quad \mu_c^s := \text{mean}\{k(z_i^s, \cdot) : i \in [n_s] \text{ and } y_i^s = c\}, \quad \forall c \in [C] \tag{15}$$

$$\mu_c^t := \text{mean}\{k(z_j^t, \cdot) : j \in [n_t] \text{ and } \hat{y}_j^t = c\}, \quad \forall c \in [C]. \tag{16}$$

Here $\beta > 0$ is a tradeoff coeffient. The underlying motivation is to align the class-wise center between source and target domains (the intra-class discrepancy), and push apart the centers of different classes (the inter-class discrepancy). Although the source-domain label is used to initialize clustering, the prediction head $h$ is not involved in $d_{\text{CDD}}$, hence not sufficiently driven by the end task.

In addition, a number of heuristics are required for CDD to perform well. Firstly, after clustering, only the target-domain examples that are close to the center are included to compute the mean $\mu_c^t$. This introduces one hyperparameter to tune. Secondly, domain specific batch-normalization is required. Finally, the bandwidth of the RBF kernel needs to be learned for *each pair* of $(c, c')$ in the implementation. To remove **all** these nuisances and formulate a *principled* optimization, we next warp CDD towards tasks based on bi-level optimization.

## 4.1 Implicit task-driven CDD

Our key insight is that the head $h^*$ in (12) constitutes a natural source of pseudo-label that is superior to clustering. Firstly, $h^*$ is uniquely determined thanks to the convexity originating from the linearity of $h$. Moreover, clustering is a "procedure" which is not amenable to differentiation despite some recent progress in reversible learning [46]. In contrast, differentiation through $h^*$ is straightforward as discussed above.

This intuition can be directly implemented by redefining the class centers in the target domain based on the $h^*$-induced soft pseudo-label for each example $z_j^t$. Recall $h^*(z_j^t)$ produces the $C$-dimensional logit (unnormalized score) for the $C$ classes, and the softmax of it yields a $C$-dimensional probability vector $p_j^t$, whose $c$-th element encodes the probability of belonging to class $c$. Accordingly, we can morph the target-domain center $\mu_c^t$ into

$$\mu_c^t(h) := \sum_{j=1}^{n_t} (p_j^t)_c \, z_j^t \Bigg/ \left( 10^{-6} + \sum_{j=1}^{n_t} (p_j^t)_c \right), \quad \text{where} \quad p_j^t = \text{softmax}(h(z_j^t)) \in \mathbb{R}^C. \tag{17}$$

Note the kernel $k$ is removed and we directly used $z_j^t$. We also added a small smoothing factor $10^{-6}$ in case all examples are unlikely to belong to class $c$. To summarize, our training objective is

$$\min_{\phi} d_{i\text{-CDD}}(\tilde{P}_\phi, \tilde{Q}_\phi) + \alpha \mathcal{R}(h^*; \tilde{P}_\phi) \tag{18}$$

$$\text{where} \quad d_{i\text{-CDD}}(\tilde{P}_\phi, \tilde{Q}_\phi) := \frac{1}{C} \sum_{c \in [C]} \left\| \mu_c^s - \mu_c^t(h^*) \right\|_{\mathcal{H}}^2 - \beta \frac{1}{C(C-1)} \sum_{c \neq c'} \left\| \mu_c^s - \mu_{c'}^s \right\|_{\mathcal{H}}^2 \tag{19}$$

$$h^* := \arg \min_{h \in \mathcal{H}} \mathcal{R}(h; \tilde{P}_\phi). \tag{20}$$

It is clearly identical to $i$-MDD in (11) except that the $d_{i\text{-MDD}}$ is replaced by $d_{i\text{-CDD}}$. Compared with $d_{\text{CDD}}$ in (14), we slightly changed the inter-class term from between source and target domains ($\mu_c^s - \mu_{c'}^t$) into within source domain only ($\mu_c^s - \mu_{c'}^s$). This simplifies optimization because the centers of the source domain do not depend on $h^*$. Meanwhile, different classes are still pushed apart in *both* domains because 1) it is enforced on the source domain, and 2) the source domain centers $\mu_c^s$ are aligned with those of the target domain $\mu_c^t(h^*)$. Backpropagation and bi-level optimization are similar to $i$-MDD, with even reduced complexity as no optimization (over $h'$) is involved in $d_{i\text{-CDD}}$.

## 4.2 Cache-augmented training

It was noted in [24] that the limited size of mini-batch may leave only a small number of examples for each class (or even none), especially when there are many classes. This hampers the computation of class means. They thus resorted to a class-aware sampling strategy where only a subset of classes are picked at each iteration, and samples are drawn only for *these* classes. This again relies on the result of clustering for the target domain, exacerbating the fallout of not backpropagating through it.

To address this issue, we followed [50, 51] by caching the latent representations $z$ in the most recent iterations via a circular queue for each class. This allows the class means to be computed more accurately, and the backpropagation is still conducted only on the current mini-batch examples.

We emphasize that our overall optimization remains principled even with cache augmentation, an observation that has not been made in literature to the best of our knowledge. Since $\phi$ is updated with a small step size and only a small number of latest iterations are cached, the continuity of the algorithm ensures that the $z$ computed from a stale $\phi$ is still close to the value if it *were* computed with the latest $\phi$. As a result, the bias of the gradient can be bounded linearly by the step size times the staleness (i.e., the length of the queue / mini-batch size). We relegate the details to Appendix C.

## 5 Related Works in Unsupervised Domain Adaptation via Feature Adaptation

Although our motivation is to develop a task-driven probability discrepancy measure while UDA is used only as an example application, we would also like to place our approach in the context of UDA literature. A detailed and recent survey is available in [52], and we will only focus on one category of methods that are most related to our approach, namely feature adaptation based methods. These methods seek feature extractors so that the source and target domains are aligned in the feature space, hence called domain-invariant feature representations [53]. Although methods may differ in whether different domains share the feature extractors in part, in whole, or none, the most prominent variation lies in the alignment metric.

Conventional probability discrepancy measures include Jensen-Shannon divergence used by GAN and DANN, and Wasserstein distance [54, 55]. The MMD in (3) has multiple variants such as multiple kernels [13] and joint MMD [56]. When the kernel is not universal, e.g., polynomial, MMD essentially compares the statistics such as the variance (second-order), and various comparison metrics have been studied [e.g., 57, 58]. CDD further accounts for the source-domain label information (but not the target-domain head) via the intra-class and inter-class distances.

Several adversarial methods try to align the domains by demoting the features' discriminative power in identifying the domain. The idea can be traced back to at least GAN, and example variants include DANN and [31, 59–61].

## 6 Experimental Results

We finally validate the implicit task-driven discrepancy by comparing $i$-MDD and $i$-CDD against state-of-the-art methods for unsupervised domain adaptation, especially MDD and CDD. Ablation studies will also be carried out to examine the influence of various components. More details on the experiment setup and results are available in Appendix D.

### 6.1 Comparison of target-domain accuracy

**Datasets.**  We adopted three public domain datasets for UDA benchmarking.

- **Office-31** [62] is a standard dataset for real-world domain adaptation. It consists of 4,652 images belonging to 31 unbalanced classes. These images are collected from three distinct domains: **A**mazon (from Amazon website), **W**ebcam (from web camera) and **D**SLR (by digital SLR camera).

- **Office-Home** [63] is a more challenging dataset for visual domain adaptation. It contains 15,500 images of daily objects in office or home environment, belonging to 65 categories. The images are sampled from four domains: **Ar**tistic images, **Cl**ip Art, **Pr**oduct images, and **R**eal-**w**orld images.

- **ImageCLEF-DA** [64] consists of images from three domains: **C**altech-256, **I**mageNet ILSVRC 2012 and **P**ascal VOC 2012. Each domain has 12 categories and each class contains 50 images.

**Baselines.** We compared our $i$-MDD and $i$-CDD with the following state-of-the-art UDA methods: Deep Adaptation Networks **(DAN)** [13], Domain Adversarial Neural Network **(DANN)** [14], Residual Transfer Network **(RTN)** [65], Joint Adaptation Networks **(JAN)** [64], the Entropy Conditioning Variant of Conditional Domain Adversarial Network **(CDAN+E)** [31], Multi-Adversarial Domain Adaptation **(MADA)** [66], Conditional Domain Adversarial Network with Batch Spectral Penalization **(BSP+CDAN)** [67], **CDD** [24] (which named it Contrastive Adaptation Network), Cluster Alignment with a Teacher with Robust Gradient Reversal **(rRevGrad+CAT)** [68], **MDD** [23], MDD with Implicit Alignment **(MDD+IA)** [69], and Adversarial Spectral Adaptation Network **(ASAN)** [70].

We also considered a variant of CDD (named **vCDD**) where $\mu_c^s - \mu_{c'}^t$ is replaced by $\mu_c^s - \mu_{c'}^s$ in source domain *only*, and the class-aware sampling in [24] is replaced by cache augmentation. This allows us to compare $i$-CDD with the exact counterpart that does not use bi-level optimization.

Additional comparisons with some state-of-the-art methods that are *not* based on feature adaptation are available in Appendix D.3.

**Implementation details.** We followed the commonly used experimental protocol for unsupervised domain adaptation from [14]. We report the average accuracy and standard deviation of five independent runs. For $i$-MDD we mainly used the hyper-parameters from [23], i.e., the margin factor $\gamma$ in (13) was chosen from $\{2, 3, 4\}$ and was kept the same for all tasks on the same dataset. For $i$-CDD, the trade-off coefficient $\beta$ between intra-class loss and inter-class loss in (14) is chosen from $\{0.1, 0.01, 0.001\}$. The cache size for each class is 30.

We implemented our methods in PyTorch. The head classifier (in both $i$-CDD and $i$-MDD) and the auxiliary classifier ($h'$ in $i$-MDD) are both 1-layer neural network with width 1024. We did not restrict MDD and CDD to single-layer $h$ or $h'$.

For optimization, we used mini-batch SGD with Nesterov momentum 0.9. The initial learning rate was 0.004, which was adjusted according to [14]. The mini-batch size is 150 for each domain. More detailed explanation of hyper-parameter selection is presented in the supplementary materials, along with the sensitivity analysis of them. ResNet-50 pretrained on ImageNet was used as the feature extractor in all methods. Since our aim is to improve the probability discrepancy measure for UDA, we employed the standard backbone ResNet-50 instead of integrating heavier-weight feature extractors, ad-hoc engineering heuristics, or generic feature improvements [e.g., 71].

**Results.** The accuracy of target-domain prediction is presented in Table 1 for Office-31, Table 2 for Office-Home, and Table 3 for ImageCLEF. Clearly $i$-CDD achieves the highest average accuracy among all methods over all datasets. As we zoom into each pair of domain, it is also either the best performer or close to the best. Secondly, by comparing vCDD with $i$-CDD and MDD with $i$-MDD, it is clear that the implicit (i.e., bi-level) formulation can significantly boost the performance upon the standard joint optimization, except $i$-MDD on ImageCLEF where it is a tie. This validates our original motivation. Thirdly, vCDD outperforms CDD on two datasets and ties on Office-31, implying that computing the inter-class discrepancy based solely on the source domain is superior to that based on both source and target domains. This makes sense because ground-truth labels are only available for the source, and the pseudo-labels for the target domain can be noisy and detrimental.

Overall, $i$-CDD is superior to $i$-MDD. This makes sense because $i$-CDD not only matches the center of each class between source and target, but also promotes the inter-class discrepancy, i.e., pushing apart the center of different classes. The latter "contrastive" component appears quite beneficial.

Finally, MDD+IA can often outperform MDD, and although $i$-MDD achieves significantly higher accuracy than MDD+IA on Office-31, it is less competitive on the other two datasets. This does not invalidate our implicit task-driven principle, and we can implicitize MDD+IA for future work.

## 6.2 Ablation study

We next examine the influence of several important components of $i$-MDD and $i$-CDD, including the cache size (queue length) in $i$-CDD, the dimensionality of hidden representation, $i$-CDD equipped with the class-aware sampling [24]. All the ablation studies were conducted on Ar:Cl in Office-Home.

Table 1: Accuracy (%) on Office-31 for unsupervised domain adaptation (based on ResNet-50)

| Method | A $\to$ W | D $\to$ W | W $\to$ D | A $\to$ D | D $\to$ A | W $\to$ A | Avg |
|---|---|---|---|---|---|---|---|
| ResNet-50 | 68.4 ± 0.2 | 96.7 ± 0.1 | 99.3 ± 0.1 | 68.9 ± 0.2 | 62.5 ± 0.3 | 60.7 ± 0.3 | 76.1 |
| DAN | 80.5 ± 0.4 | 97.1 ± 0.2 | 99.6 ± 0.1 | 78.6 ± 0.2 | 63.6 ± 0.3 | 62.8 ± 0.2 | 80.4 |
| DANN | 82.0 ± 0.4 | 96.9 ± 0.2 | 99.1 ± 0.1 | 79.7 ± 0.4 | 68.2 ± 0.4 | 67.4 ± 0.5 | 82.2 |
| RTN | 84.5 ± 0.2 | 96.8 ± 0.1 | 99.4 ± 0.1 | 77.5 ± 0.3 | 66.2 ± 0.2 | 64.8 ± 0.3 | 81.6 |
| JAN | 85.4 ± 0.3 | 97.4 ± 0.2 | 99.8 ± 0.2 | 84.7 ± 0.3 | 68.6 ± 0.3 | 70.0 ± 0.4 | 84.3 |
| CDAN+E | 94.1 ± 0.1 | 98.6 ± 0.1 | **100.0** ± 0.0 | 92.9 ± 0.2 | 71.0 ± 0.3 | 69.3 ± 0.3 | 87.7 |
| MADA | 90.0 ± 0.1 | 97.4 ± 0.1 | 99.6 ± 0.1 | 87.8 ± 0.2 | 70.3 ± 0.3 | 66.4 ± 0.3 | 85.2 |
| BSP+CDAN | 93.3 ± 0.2 | 98.2 ± 0.2 | **100.0** ± 0.0 | 93.0 ± 0.2 | 73.6 ± 0.3 | 72.6 ± 0.3 | 88.5 |
| CDD | 94.5 ± 0.3 | **99.1** ± 0.2 | 99.8 ± 0.2 | 95.0 ± 0.3 | **78.0** ± 0.3 | 77.0 ± 0.3 | 90.6 |
| rRevGrad+CAT | 94.4 ± 0.1 | 98.0 ± 0.2 | **100.0** ± 0.0 | 90.8 ± 1.8 | 72.2 ± 0.2 | 70.2 ± 0.1 | 87.6 |
| MDD | 94.5 ± 0.3 | 98.4 ± 0.1 | 100.0 ± 0.0 | 93.5 ± 0.2 | 74.6 ± 0.3 | 72.2 ± 0.1 | 88.9 |
| MDD+IA | 90.3 ± 0.2 | 98.7 ± 0.1 | 99.8 ± 0.0 | 92.1 ± 0.5 | 75.3 ± 0.2 | 74.9 ± 0.3 | 88.8 |
| ASAN | **95.6** ± 0.4 | 98.8 ± 0.2 | **100.0** ± 0.0 | 94.4 ± 0.9 | 74.7 ± 0.3 | 74.0 ± 0.9 | 90.0 |
| vCDD | 95.1 ± 0.7 | 98.4 ± 0.3 | 99.5 ± 0.3 | 94.8 ± 0.7 | 76.2 ± 0.5 | 76.9 ± 0.6 | 90.6 |
| $i$-CDD | 95.4 ± 0.4 | 98.5 ± 0.2 | **100.0** ± 0.0 | **96.3** ± 0.3 | 77.2 ± 0.3 | **78.3** ± 0.2 | **90.9** |
| $i$-MDD | 94.8 ± 0.5 | 98.4 ± 0.3 | **100.0** ± 0.0 | 94.2 ± 0.5 | 75.1 ± 0.5 | 74.1 ± 0.7 | 89.4 |

Table 2: Accuracy (%) on Office-Home for unsupervised domain adaptation (based on ResNet-50)

| Method | Ar:Cl | Ar:Pr | Ar:Rw | Cl:Ar | Cl:Pr | Cl:Rw | Pr:Ar | Pr:Cl | Pr:Rw | Rw:Ar | Rw:Cl | Rw:Pr | Avg |
|---|---|---|---|---|---|---|---|---|---|---|---|---|---|
| ResNet-50 | 34.9 | 50.0 | 58.0 | 37.4 | 41.9 | 46.2 | 38.5 | 31.2 | 60.4 | 53.9 | 41.2 | 59.9 | 46.1 |
| DAN | 43.6 | 57.0 | 67.9 | 45.8 | 56.5 | 60.4 | 44.0 | 43.6 | 67.7 | 63.1 | 51.5 | 74.3 | 56.3 |
| DANN | 45.6 | 59.3 | 70.1 | 47.0 | 58.5 | 60.9 | 46.1 | 43.7 | 68.5 | 63.2 | 51.8 | 76.8 | 57.6 |
| JAN | 45.9 | 61.2 | 68.9 | 50.4 | 59.7 | 61.0 | 45.8 | 43.4 | 70.3 | 63.9 | 52.4 | 76.8 | 58.3 |
| CDAN+E | 50.7 | 70.6 | 76.0 | 57.6 | 70.0 | 70.0 | 57.4 | 50.9 | 77.3 | 70.9 | 56.7 | 81.6 | 65.8 |
| BSP+CDAN | 52.0 | 68.6 | 76.1 | 58.0 | 70.3 | 70.2 | 58.6 | 50.2 | 77.6 | 72.2 | 59.3 | 81.9 | 66.3 |
| CDD | 51.6 | 71.2 | 76.7 | 59.8 | 70.8 | 70.8 | 59.8 | 49.9 | 77.4 | 70.6 | 58.8 | 80.5 | 66.5 |
| MDD | 54.9 | 73.7 | 77.8 | 60.0 | 71.4 | 71.8 | 61.2 | 53.6 | 78.1 | **72.5** | 60.2 | 82.3 | 68.1 |
| MDD+IA | 56.2 | **77.9** | **79.2** | **64.4** | 73.1 | **74.4** | 64.2 | 54.2 | **79.9** | 71.2 | 58.1 | 83.1 | 69.5 |
| ASAN | 53.6 | 73.0 | 77.0 | 62.1 | 73.9 | 72.6 | 61.6 | 52.8 | 79.8 | 73.3 | 60.2 | **83.6** | 68.6 |
| vCDD | 56.2 | 74.2 | 77.0 | 62.4 | 72.3 | 71.4 | 61.7 | 61.4 | 78.7 | 71.3 | 60.6 | 81.7 | 69.3 |
| $i$-CDD | **60.8** | 77.5 | 78.8 | **64.3** | **74.3** | 73.4 | **65.3** | **61.9** | 78.7 | 72.1 | **61.8** | 81.8 | **70.8** |
| $i$-MDD | 56.5 | 74.7 | 78.3 | 61.9 | 72.4 | 72.3 | 63.2 | 55.6 | 78.4 | 71.4 | 59.7 | 81.7 | 68.8 |

**Impact of cache size in $i$-CDD and vCDD.** Figure 3 shows the fluctuation of prediction accuracy for vCDD and $i$-CDD. The accuracy first grows when the length of the queue for each class increases from 1 to 30, corroborating the benefit of cache in improving the accuracy of center means. But then it starts to decay, suggesting that the stale samples accrued start to hurt.

Since there is a large number of class compared with the mini-batch size, the cluster mean cannot be estimated accurately. For example, Ar:Cl of Office-Home has 65 classes while the GPU memory limited our mini-batch size to 150. The cache augments the pool of latent feature values, hence improving the mean estimation. However, an overly large queue size may leave the stored values stale, i.e., inconsistent with the true value if it *were* computed from the current ResNet $\phi$.

Empirically, we found it generally effective to set the cache size to around 50% of the data size (number of images) of each domain. For example, there are about 2000 images in the Amazon website domain of Office-31, and we set the queue length to 30 for each of the 31 classes. This amounted to a cache size of $30 \times 31 = 930$ images, which is about half of 2000. It well balanced the sample size with staleness, and cost only a small amount of memory and computation thanks to the low dimensionality of the latent feature space.

**Impact of latent dimensionality.** Figure 4 demonstrates the prediction accuracy of vCDD and $i$-CDD, when the dimensionality of latent feature ($z_i^s$ and $z_j^t$) is varied in $\{128, 256, 512, 1024\}$.

Table 3: Accuracy (%) on ImageCLEF for unsupervised domain adaptation (based on ResNet-50)

| Method | I → P | P → I | I → C | C → I | C → P | P → C | Avg |
|---|---|---|---|---|---|---|---|
| ResNet-50 | 74.8 ± 0.3 | 83.9 ± 0.1 | 91.5 ± 0.3 | 78.0 ± 0.2 | 65.5 ± 0.3 | 91.2 ± 0.3 | 80.7 |
| DAN | 74.5 ± 0.4 | 82.2 ± 0.2 | 92.8 ± 0.2 | 86.3 ± 0.4 | 69.2 ± 0.4 | 89.8 ± 0.4 | 82.5 |
| DANN | 75.0 ± 0.6 | 86.0 ± 0.3 | 96.2 ± 0.4 | 87.0 ± 0.5 | 74.3 ± 0.5 | 91.5 ± 0.6 | 85.0 |
| RTN | 75.6 ± 0.3 | 86.8 ± 0.1 | 95.3 ± 0.1 | 86.9 ± 0.3 | 72.7 ± 0.3 | 92.2 ± 0.4 | 84.9 |
| JAN | 76.8 ± 0.4 | 88.0 ± 0.2 | 94.7 ± 0.2 | 89.5 ± 0.3 | 74.2 ± 0.3 | 91.7 ± 0.3 | 85.8 |
| CDAN+E | 77.7 ± 0.3 | 90.7 ± 0.2 | **97.7** ± 0.3 | 91.3 ± 0.3 | 74.2 ± 0.2 | 94.3 ± 0.3 | 87.7 |
| MADA | 75.0 ± 0.3 | 87.9 ± 0.2 | 96.0 ± 0.3 | 88.8 ± 0.3 | 75.2 ± 0.2 | 92.2 ± 0.3 | 85.8 |
| CDD | 77.0 ± 0.5 | 89.4 ± 0.3 | 97.2 ± 0.3 | 91.5 ± 0.2 | 76.2 ± 0.5 | 95.6 ± 0.6 | 87.8 |
| rRevGrad+CAT | 77.2 ± 0.2 | 91.0 ± 0.3 | 95.5 ± 0.3 | 91.3 ± 0.3 | 75.3 ± 0.6 | 93.6 ± 0.5 | 87.3 |
| MDD | 78.5 ± 0.2 | 91.1 ± 0.4 | 97.0 ± 0.2 | **92.1** ± 0.4 | 77.6 ± 0.3 | 93.8 ± 0.4 | 88.4 |
| MDD+IA | 78.3 ± 0.2 | 91.8 ± 0.2 | 96.7 ± 0.3 | 93.0 ± 0.2 | **79.0** ± 0.3 | 94.2 ± 0.2 | 88.8 |
| ASAN | 78.9 ± 0.4 | 92.3 ± 0.5 | 97.4 ± 0.5 | **92.1** ± 0.3 | 76.4 ± 0.7 | 94.4 ± 0.2 | 88.6 |
| vCDD | 78.8 ± 0.4 | 92.1 ± 0.1 | 97.0 ± 0.3 | 91.3 ± 0.3 | 78.2 ± 0.3 | 96.2 ± 0.4 | 88.9 |
| $i$-CDD | **79.8** ± 0.4 | **92.6** ± 0.3 | 97.2 ± 0.4 | **92.0** ± 0.3 | 78.6 ± 0.3 | **96.5** ± 0.2 | **89.4** |
| $i$-MDD | 78.5 ± 0.6 | 91.6 ± 0.5 | 96.5 ± 0.4 | 91.4 ± 0.3 | 76.8 ± 0.6 | 95.4 ± 0.3 | 88.4 |

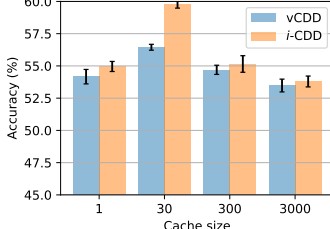

Figure 3: Accuracy v.s. cache size *for each class*

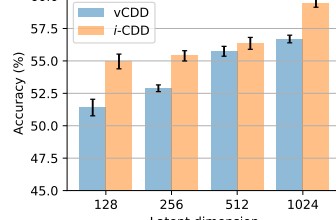

Figure 4: Accuracy v.s. latent dimensionality

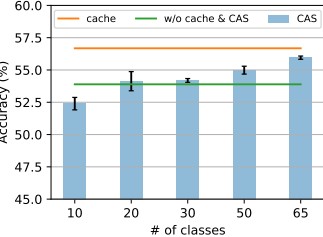

Figure 5: Class-aware sampling v.s. cache augmentation

Evidently, increasing the dimensionality tends to improve the accuracy for both methods, but at the cost of more computation.

**Class-aware sampling v.s. cache augmentation.** The problem of low sample for each class in a mini-batch (ref Section 4.2) was addressed by [24] via class-aware sampling (CAS), where a small number of classes (e.g., 10) are randomly selected, and a mini-batch only draws samples from these classes. Essentially, each iteration is based only on a subset of classes, while our $i$-CDD and vCDD still allow all classes to participate via cache augmentation. It is therefore of interest to compare CAS with cache. As shown in Figure 5, vCDD using CAS enjoys a monotonic growth of accuracy as more and more classes are involved in each iteration. When all the 65 classes are used, CAS gets close to cache augmentation. Without cache or CAS, the performance is lower (green line). This partly explains the success of vCDD, which is later improved further by $i$-CDD via the bi-level formulation.

Additional ablations studies are available in Appendix D.4, including the impact of batch size and standard deviations.

# 7    Conclusion

In this paper, we proposed warping probability discrepancy measures towards the end tasks by leveraging the pseudo-labels produced by the optimal predictor. Application to unsupervised domain adaptation significantly outperformed the state of the art in prediction accuracy, and the training is formulated as a principled optimization problem solvable by standard optimizers. For future work, it will be interesting to extend this technique to warping (conditional) independence measures, and to apply to structured and dynamic settings.

## Acknowledgements

We thank the reviewers for their constructive comments. This work is supported by NSF grant RI:1910146 and NIH grant R01CA258827.

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
