# Supplementary Material

## A    Example Comparing $d_{\text{MDD}}$ and $d_{i\text{-MDD}}$ in Conjunction with $\mathcal{R}$

We now compare $d_{\text{MDD}}(P,Q)$ and $d_{i\text{-MDD}}(P,Q)$ in the context of $\mathcal{R}$. To be self-contained, we copy their definitions from (6) and (11) to here:

$$d_{\text{MDD}}(P,Q) := \min_{h \in \mathcal{H}} \left\{ \mathcal{D}(h) + \lambda \mathcal{R}(h) \right\}, \tag{21}$$

$$\text{where} \quad \mathcal{D}(h) := \max_{h' \in \mathcal{H}} \mathcal{D}(h, h', P, Q), \qquad \mathcal{R}(h) := \mathop{\mathbb{E}}_{(z,y) \in P} \ell(h(z), y) + \text{reg}(h). \tag{22}$$

And

$$d_{i\text{-MDD}}(P,Q) := \mathcal{D}(h^*), \quad \text{where} \quad h^* := \arg\min_{h \in \mathcal{H}} \mathcal{R}(h). \tag{23}$$

Here for simplicity, we abused the symbol $\mathcal{D}$ in (22) by maximizing out $h'$ in the original $\mathcal{D}$. No confusion will arise because the input argument clearly distinguishes the meaning. We also kept the dependency on $P$ and $Q$ implicit in all terms. The tradeoff weight $\lambda$ is not the one in (10).

**Case 1**: $\lambda$ is small. In this case, $d_{\text{MDD}}$ places a low weight on fitting the source-domain data, which differs substantially from the motivation of $d_{i\text{-MDD}}$. This is obviously not a good choice, and in general, MDD does not operate in this regime.

**Case 2**: $\lambda$ is large. This appears to make $d_{\text{MDD}}$ close to $d_{i\text{-MDD}}$ because the large value of $\lambda$ will push $h$ to focus on minimizing $\mathcal{R}$, which is consistent with the definition of $h^*$ in $d_{i\text{-MDD}}$. However, with large $\lambda$, the value of $\lambda \mathcal{R}(h)$ under the optimal $h$ for $\mathcal{D}(h) + \lambda \mathcal{R}(h)$ can get very large which significantly overshadows $\mathcal{D}$, making $d_{\text{MDD}}$ overlook the discrepancy measure $\mathcal{D}$.

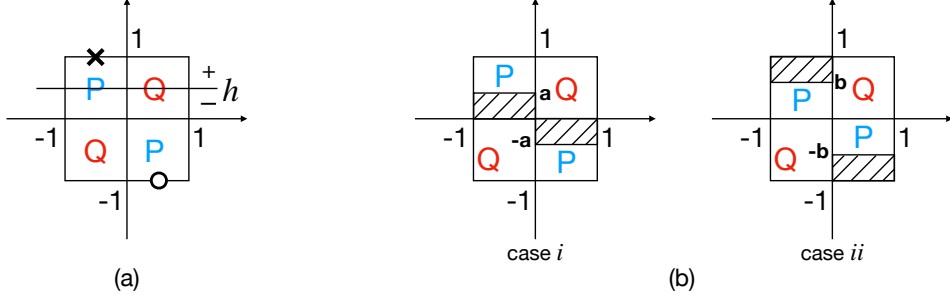

Figure 6: Examples for comparing $d_{\text{MDD}}$ and $d_{i\text{-MDD}}$. (a): for large $\lambda$. (b): for medium $\lambda$.

To see an example, consider a variant of Figure 1 where the data uniformly fills $[-1, 1] \times [-1, 1]$ as plotted in Figure 6 (a). $P$ and $Q$ are the source and target domains, respectively. In the top-left area $P$, suppose only one example (marked by x with vertical coordinate 1) is confidently labeled as positive, and the rest examples are highly inconfidently labeled, hence not to contribute to the risk $\mathcal{R}$. Similarly, there is only one confidently labeled example ($\circ$) in the bottom-right area of $P$, and it is negative with vertical coordinate $-1$. Since $h$ (as a hypothesis) shifts vertically, we will also use $h$ to denote its coordinate on the vertical axis. As was explained in the caption of Figure 1, $\mathcal{D}(h) = 1 - h$. Since the distance between $h$ and the positive x is $1 - h$, the probability of $x$ being positive, according to $h$, is sigmoid$(1 - h)$. Similarly, the probability of $\circ$ being negative, according to $h$, is $1 - \text{sigmoid}(-1 - h)$. Putting them together, we get the following $\mathcal{R}$ with cross-entropy loss and no regularization

$$\min_{h \in [0,1]} \lambda \underbrace{\left( \log(1 + e^{h-1}) + \log(1 + e^{-1-h}) \right)}_{\mathcal{R}(h)} + \underbrace{1 - h}_{\mathcal{D}(h)}. \tag{24}$$

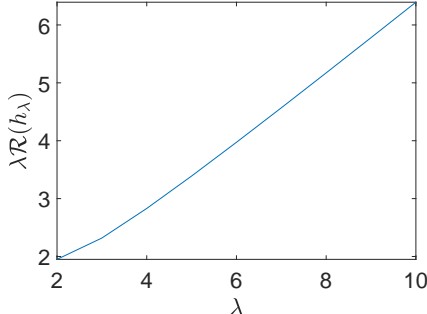

Figure 7: $\lambda \mathcal{R}(h_\lambda)$ as a function of $\lambda$

Whenever $\lambda > 2$, the optimal $h_\lambda$ is in $(0, 1)$ and can be solved by a quadratic equation. Figure 7 shows that $\lambda \mathcal{R}(h_\lambda)$ diverges linearly in $\lambda$. Flipping $h$ to $[-1, 0]$ produces the same issue.

In contrast, $d_{i\text{-MDD}}$ is immune to this problem because $\mathcal{R}$ is used only to determine $h^*$, while the $d_{i\text{-MDD}}$ value itself is solely contributed by $\mathcal{D}$. Although the $i$-MDD objective in (11) also has a coefficient $\alpha$ on $\mathcal{R}$, the optimization there is on the feature $\phi$, not on $h$ any more.

**Case 3**: $\lambda$ is medium. Here we will study two distributions as shown in Figure 6 (b), and analyze how $i$-MDD produces reasonable preferences of "better aligned distribution", and how MDD produces less justifiable preferences.

Same as the scenario of large $\lambda$, we do not change the feature distribution of source and target domains, hence keeping $\mathcal{D}(h) = 1 - |h|$. Instead, we vary the confidence of labels in the source domain in order to generate new risk $\mathcal{R}$. In case i (left of Figure 6 (b)), we activate (make the label confident) the positive examples in the top-left $P$ if, and only if, its vertical coordinate is in $[0, a]$ ($a \in [0, 1]$). Similarly, we activate the negative examples in the bottom-right $P$ if, and only if, its vertical coordinate is in $[-a, 0]$. The activated areas are shaded.

In case ii, (right of Figure 6 (b)), we activate the positive examples in the top-left $P$ if, and only if, its vertical coordinate is in $[b, 1]$ ($b \in [0, 1]$). Similarly, we activate the negative examples in the bottom-right $P$ if, and only if, its vertical coordinate is in $[-1, -b]$. The activated areas are shaded.

Adopting a tiny regularizer $\epsilon |h|$ with very small $\epsilon > 0$, it is clear that in both cases, and *regardless of the value of $a$ and $b$*, the optimal $h^*$ is 0. Therefore, the $d_{i\text{-MDD}}$ value is 1, which properly quantifies the discrepancy between $P$ and $Q$ regardless of the disclosure of source-domain labels.

However, the computation for $d_{\text{MDD}}$ is a little more involved. We first plot $\mathcal{R}(h)$ as a function of $h$ here:

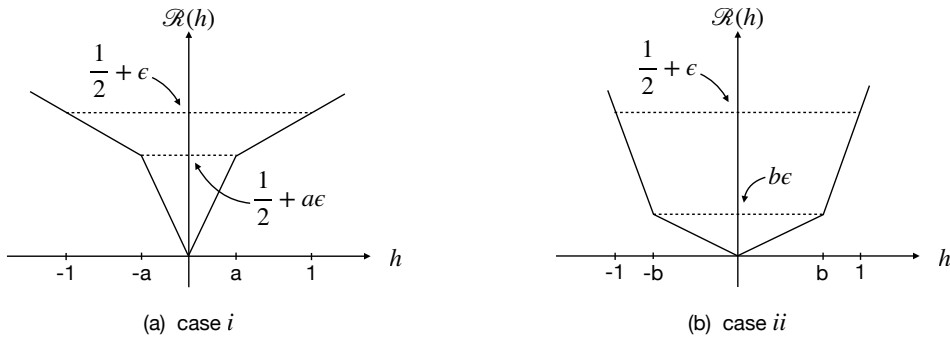

(a) case *i*          (b) case *ii*

Figure 8: Plot of $\mathcal{R}(h)$ for case i and ii in Figure 6 (b)

Then the plot of $\mathcal{D}(h) + \lambda \mathcal{R}(h)$ is

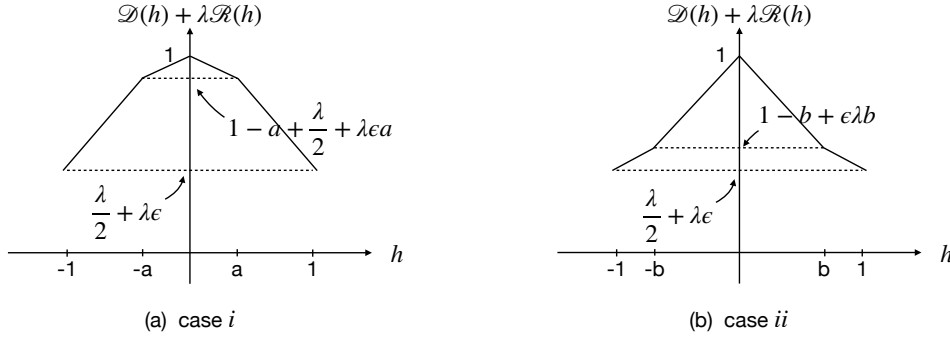

(a) case *i*                                    (b) case *ii*

Figure 9: Plot of $\mathcal{D}(h) + \lambda\mathcal{R}(h)$ for case i and ii in Figure 6 (b)

So we have

$$d_{\text{MDD}} = \begin{cases} \min\{1, \frac{\lambda}{2} + \lambda\epsilon, 1 - a + \frac{\lambda}{2} + \lambda a\epsilon\} & \text{case i} \\ \min\{1, \frac{\lambda}{2} + \lambda\epsilon, 1 - b + \lambda b\epsilon\} & \text{case ii} \end{cases}. \tag{25}$$

If $\lambda > 2$, then $d_{\text{MDD}} = 1$ for case i, while that for case ii is strictly less than 1 unless $b = 0$ ($\epsilon$ is infinitesimally small). So case ii is always preferred.

If $\lambda \le 2$, then $d_{\text{MDD}} = \frac{\lambda}{2} + \lambda\epsilon$ for case i. So there are only two situations left depending on $b$ for case ii.

- $b \in [0, 1 - \frac{\lambda}{2}]$: both cases have $d_{\text{MDD}} = \frac{\lambda}{2} + \lambda\epsilon$, i.e., equally preferred. This is the desirable outcome.

- $b \in [1 - \frac{\lambda}{2}, 1]$: then $d_{\text{MDD}} = 1 - b + \lambda b\epsilon$ for case ii, and it is therefore preferred to case i.

To summarize, $d_{\text{MDD}}$ always prefers case ii to case i, except when $\lambda < 2$ and $b \in [0, 1 - \frac{\lambda}{2}]$, in which case it is a tie. This is clearly not desirable because, by symmetry, there is no reason to prefer case ii. It is also particularly concerning that the value of $a$ in case i does not make any difference to the preference. As such, $d_{\text{MDD}}$ is not as good as a $d_{i\text{-MDD}}$ in this example.

## B   Detailed Formula for Bi-level Optimization

Let $\phi$ be the feature extractor which produces latent states $z^s := \phi(x^s)$ and $z^t := \phi(x^t)$. Let $m$ be the number of latent features, i.e., the dimensionality of $z^s$ and $z^t$. Recall $C$ is the number of classes. Denote

$$M(h, \phi) := \max_{h'} \mathcal{D}(h', h, \phi). \tag{26}$$

For convenience, we will denote the optimal $h'$ as $h'(h, \phi)$.

Given $\phi$, the $h$ can be determined by minimizing the risk on $\tilde{P}$ as in (12):

$$h_\phi := \arg\min_h \mathcal{R}(h, \phi). \tag{27}$$

Our overall optimization objective is

$$\min_\phi M(h_\phi, \phi) + \alpha\mathcal{R}(h_\phi, \phi) \Longleftrightarrow \min_\phi \left\{ M(h_\phi, \phi) + \alpha \min_h \mathcal{R}(h, \phi) \right\}. \tag{28}$$

To optimize $\phi$, we just need to compute the gradient in $\phi$. Since both $M$ and $\mathcal{R}$ depend on $\phi$ only through $z^s$ and $z^t$, we can consider the following equivalent objective

$$J(z) := M(h_z, z) + \alpha \min_h \mathcal{R}(h, z), \quad \text{where} \quad h_z := \arg\min_h \mathcal{R}(h, z). \tag{29}$$

Once the derivative $\frac{\partial J}{\partial z}$ is computed, the original derivative in $\phi$ can be easily computed through backpropagation. We will use mini-batches with size $b$.

**Step 1.** The second term in (29), $\min_h \mathcal{R}(h,z)$, admits a straightforward calculation of the derivative in $z$ thanks to the Danskin's theorem: $\nabla_z^\top \mathcal{R}(h_z, z) = \frac{\partial}{\partial z}\big|_{h_z,z} \mathcal{R}(h,z)$. Here $\nabla_z^\top$ stands for the transpose of the gradient in $z$ — hence a row vector — of $\mathcal{R}(h_z, z)$ (regarded as a function of $z$ only).

**Step 2.** The first term in (29), $M(h_z, z)$, poses the most challenge due to the bi-level optimization, and we can address it by using the techniques in [45]. Firstly, Eq 3 therein allows us to write

$$\nabla_z^\top M(h_z, z) = \underbrace{\frac{\partial}{\partial z}\Big|_{h_z,z} M(h,z)}_{:=(a)} \quad - \quad \underbrace{v^\top \times \frac{\partial^2}{\partial h \partial z^\top}\Big|_{h_z,z} \mathcal{R}(h,z)}_{:=(b)} \tag{30}$$

$$\text{where} \quad v^\top = \frac{\partial}{\partial h}\Big|_{h_z,z} M(h,z) \quad \times \quad \left[\frac{\partial^2}{\partial h \partial h^\top}\Big|_{h_z,z} \mathcal{R}(h,z)\right]^{-1}. \tag{31}$$

We will next show how to compute them in analytic forms, i.e., with no autodiff.

**Step 2a.** Here $(a)$ is easy to compute: first find $h'(h_z, z)$ and then $(a) = \frac{\partial}{\partial z}\mathcal{D}(h', h, z)$ evaluated at $(h'(h_z, z), h_z, z)$.

**Step 2b.** $v$ can be computed by using Algorithm 2-3 in [45]. Note in our work, $h$ is a linear classifier with a weight matrix $W \in \mathbb{R}^{m \times C}$. Accordingly, the $v$ is indeed a matrix $V \in \mathbb{R}^{m \times C}$.

Akin to Step 2a, $\frac{\partial}{\partial W}\big|_{W_z,z} M(W,z) = \frac{\partial}{\partial h}\mathcal{D}(h', W, z)$ evaluated at $(h'(W_z, z), W_z, z)$. The matrix inversion in (31) is a major obstacle, and we instantiate Algorithm 2-3 in [45] as follows:

1. Initialize by $V = D = \frac{\partial}{\partial W}\big|_{W_z,z} M(W,z) \in \mathbb{R}^{m \times C}$.

2. **for** $j = 1, \ldots,$ #max-iter **do**

3. $\quad D = D - \alpha \cdot \frac{\partial^2}{\partial W \partial W^\top}\Big|_{W_z,z} \mathcal{R}(W,z) \cdot D$

4. $\quad V = V - D$

So the computational bottleneck is step 3. However, there is a closed form to the directional Hessian if we use the cross-entropy loss, i.e.,

$$\mathcal{R}(W,z) = \mathbb{E}_{z^s \sim \tilde{P}}[-W_{:,y^s}^\top z^s + G(W^\top z^s)]. \tag{32}$$

Indeed, let

$$p^s := \frac{1}{\exp(G(W^\top z^s))} \begin{pmatrix} \exp(W_{:1}^\top z^s) \\ \vdots \\ \exp(W_{:C}^\top z^s)) \end{pmatrix}, \quad \text{where} \quad G(u) := \log \sum_{c=1}^{C} \exp(u_c). \tag{33}$$

and Appendix D of [49] shows that with $\mathbf{1}_C = (1, \ldots, 1)^\top \in \mathbb{R}^C$, $\tilde{P}(x^s) = \frac{1}{b}$, $P = (p^1, \ldots, p^b)$, $Z = (z^1, \ldots, z^b)$,

$$\frac{\partial^2}{\partial W \partial W^\top}\Big|_{W_z,z} \mathcal{R}(W,z) \cdot D = \frac{1}{b} Z \left[Q^\top - P^\top \circ (\mathbf{1}_C^\top \otimes (Q^\top \mathbf{1}_C))\right], \tag{34}$$

$$\text{where} \quad Q = P \circ (D^\top Z). \tag{35}$$

Here $\otimes$ is the Kronecker product, and $\circ$ is the Hadamard product (elementwise). Since only $S$ changes over the iterations on $j$ while $Z$ does not, we can pre-compute $P$ and $Z^\top Z$. Furthermore, we only need to compute the $Q^\top - P^\top \circ (\mathbf{1}^\top \otimes (Q^\top \mathbf{1}))$ as a surrogate for $D$, and then use the pre-computed $Z^\top Z$ in $Q$.

**Step 2c.** Given $v$, we will compute $(b)$ as follows. Since $W$ is a matrix, the derivative can be complicated. So we resort to the vectorization operator $\mathbf{w} := \text{vec}(W)$, and accordingly, we can consider $v$ as the vectorization of a matrix $V \in \mathbb{R}^{m \times C}$. Then the derivative in $\mathbf{w}$ can be written as

$$\frac{\partial}{\partial \mathbf{w}} \mathcal{R}(\mathbf{w}, z) = \mathbb{E}_{z^s \sim \tilde{P}}[(p^s - e_{y^s}) \otimes z^s], \tag{36}$$

where $e_{y^s}$ is the $y^s$-th canonical vector in $\mathbb{R}^C$. We next compute $v^\top \frac{\partial^2}{\partial \mathbf{w} \partial z^\top} \mathcal{R}(\mathbf{w}, z)$.

Obviously its derivative in $z^t$ is 0, and its derivative in $z^s$ is

$$\tilde{P}(x^s)v^\top \frac{\partial}{\partial z^s}[(p^s - e_{y^s}) \otimes z^s] = \tilde{P}(x^s)v^\top (\frac{\partial}{\partial z^s}[p^s \otimes z^s] - e_{y^s} \otimes I_m) \tag{37}$$

$$= \tilde{P}(x^s)v^\top \frac{\partial}{\partial z^s}[p^s \otimes z^s] - \tilde{P}(x^s)V_{:,y^s}^\top, \tag{38}$$

where $I_m \in \mathbb{R}^{m \times m}$ is the identity matrix and $v = \text{vec}(V)$. To compute the first term in (38), we drop the superscript $s$ for simplicity. Notice that for any class $c$ from 1 to $C$, we have

$$\frac{\partial p_c}{\partial z} = p_c W_{:c}^\top - p_c \sum_{i=1}^{C} p_i W_{:i}^\top = p_c(e_c - p)^\top W^\top. \tag{39}$$

Therefore

$$\frac{\partial}{\partial z}(p_c z) = p_c I_m - z \frac{\partial}{\partial z} p_c = p_c(I_m - z(e_c - p)^\top W^\top), \tag{40}$$

which implies that

$$v^\top \frac{\partial}{\partial z}[p \otimes z] = \sum_c p_c V_{:c}^\top (I_m - z(e_c - p)^\top W^\top) \tag{41}$$

$$= (Vp)^\top + (z^\top Vp)(Wp)^\top - [p^\top \circ (z^\top V)]W^\top. \tag{42}$$

This can be computed efficiently because it only involves matrix-vector multiplication. In practice, we would like to do it in a batch for all $s$ (recall we have dropped this superscript). Letting $A = VP$, $B = WP$, $F = Z^\top V$, it is not hard to derive that

$$\begin{pmatrix} v^\top \frac{\partial}{\partial z^1}[p^1 \otimes z^1] \\ v^\top \frac{\partial}{\partial z^2}[p^2 \otimes z^2] \\ \vdots \end{pmatrix} = A^\top + [(A^\top \circ Z^\top)\mathbf{1}_m \mathbf{1}_m^\top] \circ B^\top - (P^\top \circ F)W^\top. \tag{43}$$

To construct $(V_{:,y^1}, \ldots, V_{:,y^b})^\top$, we can utilize the infrastructure in the programming language. For example, in MATLAB, it can be easily computed by $V(:, [y^1, \ldots, y^b])'$.

## B.1 Analysis of computational cost

The calculation of the derivatives of the second term in (29) and the part (a) in (30) is straightforward. The computational cost is $\mathcal{O}(bm)$. Recall that the inverse Hessian vector production in (31) is the main computational bottleneck. The approximation algorithm in Step 2b can be solved with $\mathcal{O}(i_{\max}bmC)$, where $i_{\max}$ is the number of maximal iterations. The matrix vector multiplication in Step 2c costs $\mathcal{O}(bmC)$. Therefore, the total computational cost is upper bounded by $\mathcal{O}(i_{\max}bmC)$.

In practice, we used conjugate gradient (CG), where the $i_{\max}$ stands for the maximum number of iterations for CG. We set $m = 1024$, $b = 150$, and $C$ can be at most 65 in our datasets. Instead of limiting the maximum number of iterations, we set the tolerance of convergence to $10^{-5}$. The final time cost for completing CG over the entire mini-batch was less than a second, and the remaining operations in implicit differentiation (30) are much less expensive.

## C  Bounding the gap in gradient from cache augmentation

The key advantage of $i$-CDD is the principled optimization. While the cache augmentation in Section 4.2 may appear ad hoc, we point out here that it only introduces a bias in the gradient optimization that can be bounded linearly by the queue length, i.e., staleness.

Suppose our mini-batch size is $b$ and the input samples drawn at iteration $\tau$ are $\{x_i^\tau\}_{i=1}^b$. Note we do not distinguish source or target domain and simply treat them as $x_i^\tau$. Suppose at the beginning of iteration $\tau$, the feature extractor is $\phi_\tau$. Then the latent features are $z_i^\tau = \phi_\tau(x_i^\tau)$. Suppose we store the latent feature of the past $s$ steps, i.e., $\{z_i^{\tau-1}\} \cup \ldots \cup \{z_i^{\tau-s}\}$. That is, $s$ is our staleness factor. To simplify notation, we denote $z^{\tau-1} := \{z_i^{\tau-1}\}$ and $z^{\tau_1:\tau_2} := z^{\tau_1} \cup \ldots \cup z^{\tau_2}$. Suppose the ultimate objective value of $i$-CDD is $J$, then our algorithm with cache augmentation computes the gradient in $\phi$ at iteration $\tau$ as

$$g := \frac{1}{b} \sum_{i=1}^{b} \frac{\partial z_i^\tau}{\partial \phi} \frac{\partial}{\partial z_i^\tau} J(z^{\tau-s:\tau}). \tag{44}$$

Here the average is only on the $z_i^\tau$ of the current iteration $\tau$, although $J$ is computed using stale $z$ features in $\tau - 1, \ldots, \tau - s$.

Our goal is to bound the distance between $g$ and the "correctly" computed gradient. It is important to note that $z_i^{\tau-1}$ is computed by the *past* features $\phi_{\tau-1}$, not the current $\phi_\tau$. Hypothetically, if we could compute them by using the latest $\phi_\tau$, then let us denote such fictitious $z$ as $\hat{z}_i^{\tau-1} := \phi_\tau(x_i^{\tau-1})$ and define a syntactic sugar $\hat{z}_i^\tau = z_i^\tau$. Then the "correct" gradient from a principled stochastic gradient can be computed by

$$g^* := \frac{1}{b(s+1)} \sum_{\pi=\tau-s}^{\tau} \sum_{i=1}^{b} \frac{\partial \hat{z}_i^\pi}{\partial \phi} \frac{\partial}{\partial \hat{z}_i^\pi} J(\hat{z}^{\tau-s:\tau}). \tag{45}$$

So we can bound the bias by

$$\|g - g^*\| \le \|g - \hat{g}\| + \|\hat{g} - g^*\|, \quad \text{where} \quad \hat{g} := \frac{1}{b} \sum_{i=1}^{b} \frac{\partial z_i^\tau}{\partial \phi} \frac{\partial}{\partial z_i^\tau} J(\hat{z}^{\tau-s:\tau}). \tag{46}$$

Firstly, $g^*$ and $\hat{g}$ both evaluate $J$ based on the augmented sample $\hat{z}^{\tau-s:\tau}$ that is computed hypothetically through the latest $\phi_t$. The former then averages the partial derivative over all the $b(s+1)$ samples while the latter only averages over the latest $b$ samples. This deviation does *not* involve any staleness, and can be bounded by standard concentration bounds such as Hoeffding's inequality.

Secondly, $g$ and $\hat{g}$ differ only in how $J$ is computed. The former uses the stale samples $z^{\tau-s:\tau}$, while the latter uses the fictitious samples $\hat{z}^{\tau-s:\tau}$. Since $J$ is a smooth function,

$$\frac{\partial}{\partial z_i^\tau} J(z^{\tau-s:\tau}) - \frac{\partial}{\partial z_i^\tau} J(\hat{z}^{\tau-s:\tau}) \tag{47}$$

can be bounded by the difference in the input arguments of $J$. Since the gradient in $\phi$ is bounded, so $\|\phi_\tau - \phi_{\tau-s}\| \le \mathcal{O}(s)$. Therefore, $\|z_i^{\tau-s} - \hat{z}_i^{\tau-s}\| \le \mathcal{O}(s)$, and the mean averaging inside $J$ implies

$$\left\| \frac{\partial}{\partial z_i^\tau} J(z^{\tau-s:\tau}) - \frac{\partial}{\partial z_i^\tau} J(\hat{z}^{\tau-s:\tau}) \right\| \le \mathcal{O}(s) \quad \text{and hence} \quad \|g - \hat{g}\| \le \mathcal{O}(s). \tag{48}$$

To summarize, the error in the gradient consists of the standard stochastic gradient noise, along with a term that is bounded linearly by the staleness $s$, which is in turn linear in the cache/queue size. So as long as we do not keep many past features, the optimization will work well. An empirical study has been shown in Section 6.2, and the values of $b$ and cache size have been provided there.

# D    Experiment Details

## D.1    Implementation details

We used the official code of CDD, MDD, and MDD+IA to produce the results for Office-Home and Image-CLEF datasets. For other baselines, since the experimental configurations are the same, we quoted the highest results in the corresponding literature. Our PyTorch implementaion is available at https://www.dropbox.com/sh/8e2enu3mwl7oxwk/AAAT8_xqkyLzLMqxqFH6tTjWa?dl=0.

We first implemented a variant of CDD, named vCDD, where $\mu_c^s - \mu_{c'}^t$ was replaced by $\mu_c^s - \mu_{c'}^s$ in source domain *only*, and the class-aware sampling in [24] was replaced by cache augmentation. This allowed us to compare $i$-CDD with the exact counterpart that does not use bi-level optimization. We used ResNet-50 pre-trained on ImageNet as the feature extractor of vCDD model. The last FC layer of ResNet-50 was replaced by a 2-layer bottleneck neural network, where each layer has 1024 hidden units and batch normalization and sigmoid activation were applied to the hidden outputs. The bottleneck was immediately followed by a 1-layer classifier with multiple softmax units, each of which corresponds to an output class. $i$-CDD model used the same network architecture.

For $i$-MDD, to make a fair comparison, we followed MDD [23] to implement the network. ResNet-50 was adopted as the feature extractor with parameters pre-trained on ImageNet. The last FC layer of ResNet-50 was replaced by a 1-layer bottleneck network, where batch normalization, ReLU activation, and Dropout were applied to the outputs of the 1024 hidden units. Since we expected that a simple linear classifier could achieve high accuracy on the latent representations, instead of using 2-layer neural network, the main classifier $h$ and auxiliary classifier $h'$ were 1-layer neural network with width 1024.

### D.2 Hyper-parameter selection

Each method has hyper-parameters that are selected using the validation set which is comprised of labeled source examples and unlabeled target examples. The dimensionality of latent representations that are used for computing disparity discrepancy objectives, e.g. $d_{i\text{-MDD}}, d_{i\text{-CDD}}$, was selected from $\{128, 256, 512, 1024, 2048\}$. The size of the circular queue (cache) for each class was selected from $\{10, 30, 50, 100, 200\}$. For $i$-MDD method, the trade-off parameter $\alpha$ in (11) was selected from $\{0.01, 0.1, 1, 10, 100\}$; the trade-off parameter $\gamma$ in (13) was selected from $\{2, 3, 4, 5, 10\}$. For CDD and $i$-CDD methods, the trade-off parameter $\beta$ in (14) and (20) was selected from $\{0.001, 0.01, 0.1, 1\}$.

The hyper-parameters that were used for producing the results are summarized here:

Table 4: Hyper-parameters for all algorithms

| Dataset | Algorithm | latent dimension | cache size | $\alpha$ | $\beta$ | $\gamma$ |
|---------|-----------|------------------|------------|----------|---------|----------|
| Office-31 | CDD | 1024 | 30 | - | 0.001 | - |
| | $i$-CDD | 1024 | 30 | - | 0.001 | - |
| | $i$-MDD | 1024 | - | 10 | - | 4 |
| Office-Home | CDD | 1024 | 50 | - | 0.01 | - |
| | $i$-CDD | 1024 | 50 | - | 0.01 | - |
| | $i$-MDD | 1024 | - | 10 | - | 4 |
| Image-CLEF | CDD | 2048 | 30 | - | 0.001 | - |
| | $i$-CDD | 2048 | 30 | - | 0.001 | - |
| | $i$-MDD | 2048 | - | 10 | - | 4 |

### D.3 Additional comparison with methods not based on feature adaptation

We also compared with three state-of-the-art methods for unsupervised domain adaptation that are not based on feature adaptation. These include [72], [73], and [74]. The performance on all the datasets is summarized in Table 5, in comparison with $i$-CDD:

Table 5: Accuracy on target domain

| Method | Office-31 | Office-Home | ImageCLEF |
|--------|-----------|-------------|-----------|
| [72] | 88.6 | 71.8 | 88.5 |
| [73] | 89.6 | 71.0 | 90.3 |
| [74] | 88.8 | 69.2 | 90.2 |
| $i$-CDD | 90.9 | 70.8 | 89.4 |

In Table 5, we conducted the experiment for [72] on ImageCLEF, and the results for each domain are as follows:

| I -> P | P -> I | I -> C | C -> I | C -> P | P -> C |
|--------|--------|--------|--------|--------|--------|
| $77.4 \pm 0.5$ | $92.2 \pm 0.6$ | $96.1 \pm 0.2$ | $91.7 \pm 0.4$ | $77.6 \pm 0.6$ | $95.8 \pm 0.4$ |

The rest of the results in the table are quoted from the original paper, after checking manually on the data and their code.

Our $i$-CDD outperforms all these methods on Office-31. In addition, [72] is inferior to $i$-CDD on ImageCLEF, and [74] is inferior on Office-Home. [73] is almost the same as $i$-CDD on Office-Home. In addition, [73] requires solving a large generalized eigenvalue systems in their Eq 7. According to their Section "Computational Complexity", the cost is $O(d_1(d_1^2 + n^2))$ for $n$ images in the source and target domains combined, and $d_1$ can be as large as 1024. So it is highly intensive in computation for large $n$. Although stochastic PCA could be applied, its impact on the performance remains unclear.

To conclude, our $i$-CDD performs very competitively overall, and it could be overly demanding to require a method outperform state of the art on *all* datasets.

### D.4 Additional ablation studies

**Impact of Batch Size**

In our methods, random sampling was used to produce mini-batch data. Obviously, the mini-batch size determines the sampling distribution of the label space. For instance, when the mini-batch size is small, it may happen that within a given batch of samples, all source samples were drawn from 10 classes among 65 classes and all target samples were drawn from another 10 classes. The class-wise alignment objectives would suffer from this between-domain class distribution shift in the form of misalignment. Therefore, we investigated the impact of batch size.

Table 6: Impact of mini-batch size on target domain accuracy (Ar → Cl, Office-Home)

| batch size | vCDD | $i$-CDD |
|---|---|---|
| 16 | 28.4 | 29.5 |
| 32 | 39.3 | 38.8 |
| 64 | 55.9 | 57.3 |
| 128 | 56.9 | 59.4 |
| 256 | 56.7 | 59.2 |

As shown in Table 6, both vCDD and $i$-CDD enjoyed performance improvement with increased mini-batch size. Both methods worked better with a larger mini-batch size. This is because large mini-batch increases the empirical class diversity in each batch. This result suggests that class-conditioned domain adaptation approaches work well when the class diversity is high, e.g., when each mini-batch covers the whole label space.

**Standard deviations of Office-Home**

To complement Table 2, we next present the mean and standard deviation of target domain accuracy for vCDD, $i$-CDD, and $i$-MDD on the Office-Home dataset. Most existing literature does not report standard deviation on this dataset, so it was not reported in Table 2.

Table 7: Accuracy (%) on Office-Home for unsupervised domain adaptation

| Method | vCDD | $i$-CDD | $i$-MDD |
|---|---|---|---|
| Ar → Cl | $56.2 \pm 0.6$ | $60.8 \pm 0.7$ | $56.5 \pm 0.5$ |
| Ar → Pr | $74.2 \pm 0.4$ | $77.5 \pm 0.7$ | $74.7 \pm 0.6$ |
| Ar → Rw | $77.0 \pm 0.6$ | $78.8 \pm 0.5$ | $78.3 \pm 0.3$ |
| Cl → Ar | $62.4 \pm 0.4$ | $64.3 \pm 0.5$ | $61.9 \pm 0.4$ |
| Cl → Pr | $72.3 \pm 0.5$ | $74.3 \pm 0.6$ | $72.4 \pm 0.4$ |
| Cl → Rw | $71.4 \pm 0.4$ | $73.4 \pm 0.5$ | $72.3 \pm 0.6$ |
| Pr → Ar | $61.7 \pm 0.7$ | $65.3 \pm 0.8$ | $63.2 \pm 0.7$ |
| Pr → Cl | $61.4 \pm 0.6$ | $61.9 \pm 0.6$ | $55.6 \pm 0.5$ |
| Pr → Rw | $78.7 \pm 0.6$ | $78.7 \pm 0.5$ | $78.4 \pm 0.3$ |
| Rw → Ar | $71.3 \pm 0.4$ | $72.1 \pm 0.5$ | $71.4 \pm 0.4$ |
| Rw → Pr | $60.6 \pm 0.5$ | $61.8 \pm 0.4$ | $59.7 \pm 0.2$ |
| Rw → Cl | $81.7 \pm 0.4$ | $81.8 \pm 0.6$ | $81.7 \pm 0.5$ |
| Avg | 69.3 | 70.8 | 68.8 |