# OpenReview forum: "Implicit Task-Driven Probability Discrepancy Measure for Unsupervised Domain Adaptation"
_NeurIPS.cc/2021/Conference — NeurIPS 2021 Poster_

### Official Review · Reviewer_PZK7 · 2021-07-15

**Rating:** 6
**Confidence:** 3

**Summary:**

In this paper, the authors propose a new bi-level optimization-based approach to handle the issue of [23], i.e., the proposed margin disparity discrepancy (MDD) measure conflicts with the H\DeltaH-divergence. The theoretical results are then applied to MDD and CDD, which gives the implicit task-driven discrepancy method, i-MDD and i-CDD. Experimental studies are done to verify the effectiveness of the proposed i-MDD and i-CDD.

**Limitations And Societal Impact:**

An explicit limitation section is needed.

**Main Review:**

Here are some comments:

Pros:
(1)	The paper is well-written and easy to follow.

(2)	The theoretical analyses are given to motivate the implicit task-driven disparity discrepancy.

(3)	New implicit task-driven MDD and CDD, i.e., i-MDD and i-CDD, are given.

Cons:

(1)	The paper needs a thorough related work section to elaborate the existing domain adaptation learning theory as well as the corresponding domain adaptation algorithms. Discussions on how the current work differentiates from existing works are needed.

(2)	Regarding Eq. (4), it is necessary to provide more details on how this equation relates to what equation of [16].

(3)	The proposed two method i-CDD and i-MDD are not well compared, e.g., pros and cons. Moreover, is there any guide on which one should be used in what scenario?

(4)	The empirical results of i-CDD and i-MDD are only comparable with the state-of-the-art methods (in each dataset, we can find some comparable baselines), which makes the methods less convincing.

(5)	Figure 3 shows that the cache size is an important factor affecting the final performance. Is there any rule to set such a hyper-parameter when facing a new UDA task? Note that different settings are used in the office 31 and office-home dataset.

Overall, this is an interesting paper with strong theoretical analyses but weak empirical results.

After rebuttal.

Thanks for the authors' response. After reading it, I would like to keep my score.



**Time Spent Reviewing:**

5

---

> ### Author Response · Authors · 2021-08-09
> **Response to various comments**
>
> We are very grateful for your detailed review.  We hope the information below helpful and will incorporate them into the paper.
>
> 1\. We will add a section of Related Work on UDA.  Our motivation was to develop a task-driven probability discrepancy measure, using UDA as an example of application.  In retrospect, a detailed survey of UDA methods does seem helpful.  That said, we have indeed discussed in the bulk of introduction (lines 37 to 72) how our work differentiates from existing works. UDA has two major categories of approach [29]: feature adaptation and self-training.
> Our approach falls into the former, where MDD and CDD represent the state of the art.  We improved upon MDD with a bi-level optimization to resolve its conflict with the $\mathcal{H} \Delta \mathcal{H}$-divergence, and extended the strategy to improve CDD as well.
> A detailed discussion on the intellectual merit is also available in our response to Reviewer iSrE.
>
>
> 2\. The expression of $\mathcal{H} \Delta \mathcal{H}$-divergence in Eq 4 can be found in the proof of Lemma 3 in [16].  It is also written in Eq 6 of [23].
>
> 3\. Our experiment shows i-CDD is generally superior to i-MDD.  This makes sense because i-CDD not only matches the center of each class between source and target, but also promotes the inter-class discrepancy, i.e., pushing apart the center of different classes.  The latter "contrastive" component appears quite beneficial.
>
> 4\. Overall, our i-CDD performs very competitively. More discussions on experimental results are available in our response to Reviewer iSrE.
>
> 5\. Since there is a large number of class compared with the mini-batch size, the cluster mean cannot be estimated accurately.  For example, Ar:Cl of Office-Home has 65 classes while the GPU memory limited our mini-batch size to 150.
> The cache augments the pool of latent feature values, hence improving the mean estimation.
> However, an overly large queue size may leave  the stored values stale,
> i.e., inconsistent with the true value if it *were* computed from the current ResNet $\phi$.
>
> Empirically, we found it generally effective to set the cache size to around 50\% of the data size (number of images) of each domain. For example, there are about 2000 images in the Amazon website domain of Office-31, and we set the queue length to 30 for each of the 31 classes.  This amounted to a cache size of $30 \times 31=930$ images, which is about half of 2000. It well balanced the sample size with staleness,
> and cost only a small amount of memory and computation thanks to the low dimensionality of the latent feature space.

---

### Official Review · Reviewer_dRho · 2021-07-16

**Rating:** 6
**Confidence:** 3

**Summary:**

The paper "Implicit Task-Driven Probability Discrepancy Measure for Unsupervised Domain Adaptation" proposes a probability discrepancy measure in the context of unsupervised domain adaptation while keeping the end goal, namely, minimizing the target domain risk, as part of the measure. The proposed measure, i-MDD, is based on a modification of the $\mathcal{H} \Delta \mathcal{H}$-divergence of Ben-David et al. (2010) and Marginal Disparity Discrepancy (MDD) of Zhang et al. (2019). The paper then discusses practical considerations like how to actually minimize the i-MDD, which is noted to be simplified due to the structure of i-MDD, and then finally discusses experiments.

**Main Review:**

This paper is a mixed bag, it starts great by clearly introducing the problem and the preliminaries, but then completely loses its momentum by muddling its presentation on the new proposed discrepancy measure, arguably the part without which I cannot recommend accepting this paper. Let me expound on this.

For someone well-versed with the jargon of domain adaptation, reading the first two sections, introduction and preliminaries, and some parts of other sections leaves the reader with a clear picture of the problem and the literature. I am not an expert in domain adaptation, and some phrases like "margin disparity discrepancy" (MDD) or "contrastive domain discrepancy" were new to me. However, I could still get a reasonably complete picture by reading the aforementioned parts of the paper or by going through some of the references cited. But this is where the good parts end, and the problems start.

The paper fails to motivate or justify its development of the proposed discrepancy measure, namely equations (11) and (12). After introducing two other relevant measures of discrepancy, $\mathcal{H} \Delta \mathcal{H}$-divergence and MDD, the paper tries to reconcile these two measures by changing the MDD to make it look like $\mathcal{H} \Delta \mathcal{H}$-divergence. I don't see the point of this exercise at all. Why is this "conflict of spirit", as the authors put it, undesirable? The whole point of minimisation over $h$ in MDD is to select for those $h \in \mathcal{H}$ that have low source domain risk. I understand that optimizing for MDD or $\mathcal{H} \Delta \mathcal{H}$-divergence isn't easy and this is where i-MDD shines, but to me this analysis of reconciliation seems forced. Next, the lines 136-143 feel disconnected. A formulation is proposed in equation (10) which intuitively seems to strike a balance between $\mathcal{H} \Delta \mathcal{H}$-divergence and MDD, but then it is changed to something very different just on the grounds that GEM doesn't work here? Finally, there's the issue of the formulation of MDD in equation (6). It doesn't seem to correspond to the one discussed in the referenced paper by Zhang et al. (2019). I would appreciate if the authors could help me see how their formulation in equation (6) relates to Definition 3.2 of Zhang et al. where they propose the MDD. Relatedly, the formulation of i-MDD seems exactly the Disparity Discrepancy (DD) discussed in Definition 3.1 of Zhang et al. except with an extra $\alpha \mathcal{R}(h^*;\tilde{P}_\phi)$ term and the fact that a specific $h$ is being used, namely $h^*$. This reinforces my point that all the reconciliation seems superfluous.

There are other issues of notation as well:

1. One line 94, it is stated that $P := \phi \\# S$ and $\tilde{P} := \phi \\# \tilde{S}$. Did the authors intend $P := \phi \\# S_x$ and $\tilde{P} := \phi \\# \tilde{S}_x$ because otherwise it doesn't make sense. The feature extractor $\phi$ maps $\mathcal{X}$ to some latent space and $S$ is a distribution on $\mathcal{X} \times \mathcal{Y}$, and so $S \circ \phi^{-1}$ does not make sense.
2. What is $Q$ on line 100? Is it $\phi \\# T_x$?
3. On line 83, instead of $\tilde{T}$ did you intend $\tilde{T}_x$?

I did not check the experimental results.

**Time Spent Reviewing:**

10 hours

---

> ### Author Response · Authors · 2021-08-07
> **Clarification on motivation and notation**
>
> We appreciate the reviewer's effort (10 hours) and the detailed review.  We understand that the major concern is the motivation, and we hope to clarify it as follows.
>
> 1\. We claimed that the MDD in Zhang et al. (2019) essentially employs the $d_{\text{MDD}}$ in our Eq 6. This can be directly observed from their Eq 24.  The second line therein has a typo where the $\max_{f'}$ should be $\mathrm{arg}\mathop{\mathrm{max}}_{f'}$.  Plugging it into the first line, Eq 24 minimizes the following expression over $\psi$:
>
> $\min_f$ { $err_P(f)$ + $\max_{f'}$ ( $disp_Q(f', f) - disp_P(f', f))$ }
>
> (we simplified the notation by omitting unimportant sub/sup-scripts).  This is exactly our Eq 6 modulo reg($h$), where $h$ and $h'$ correspond to $f$ and  $f'$, respectively.
>
> 2\. It is straightforward to proceed from Eq 10 to Eq 11-12: set $\lambda$ to $\min_h \mathcal{R}(h; P)$, i.e., the lowest cap of loss achievable over all $h \in \mathcal{H}$ (line 141: "sets $h$ to the optimal one for the source domain").  Under this $\lambda$, the optimal objective value in Eq 10 is exactly equal to the $d_{i\text{-MDD}}$ in Eq 11. One might wonder why the $max$ over $h$ in Eq 10 is turned into $min$ over $h$ in Eq 12.  This is because $\lambda$ is set to $\min_h \mathcal{R}(h; P)$.  Analogously, maximizing $f(x,y)$ over $(x-1)^2 \le 0$ is equivalent to evaluating $f(1, y)$, because $0$ is the minimum of $(x-1)^2$ attained at $x=1$.  Or to bear more resemblance to Eq 11-12, it is equal to $f(x^*, y)$ where $x^* = \text{argmin}_x \ (x-1)^2$.
>
>
> Lines 139-140 explained the motivation of taking this step: Eq 10 does not admit efficient differentiation for $\phi$ and GEM solves Eq 10 by linear approximation.  Turning into Eq 11-12 allows the problem to be solved exactly and is amenable to differentiation.  Such a choice of $\lambda$ also makes good sense.
>
> 3\. Why is this "conflict of spirit" undesirable?  The motivation of i-MDD is far from just making MDD easier to solve.  In fact, as was detailed in Section 3.1 along with the example in Figure 1, the major problem of MDD lies in the $h$ in Eq 6, which besides reducing the source domain risk, seeks to **minimize** $\max_{h'} D(h, h', P, Q)$ over $h$.  Note that $h$ and $h'$ serve to maximally reveal the discrepancy between $P$ and $Q$, which in turn guides the feature extractor $\phi$ in aligning the latent representations of the two domains.  To this end, one needs to **maximize** $D(h, h', P, Q)$ over **both** $h$ and $h'$, which is formulated by the $\mathcal{H} \Delta \mathcal{H}$-divergence in Eq 4-5. This is what we mean by "conflict of spirit", because MDD minimizes over $h$ instead of maximizing, hence failing to reveal the discrepancy between $P$ and $Q$.  This is exactly shown in Section 3.1 with illustrations in Figure 1.  More discussions in conjunction with the source domain risk are available in Appendix A.
>
> It is our key contribution to fix this issue in MDD.  The solution based on bi-level optimization is highly nontrivial, because simply flipping signs and min/max in MDD cannot help.  As such, we respectfully disagree with the comment that our "reconciliation seems forced'' and "superfluous''.  In addition, $i$-MDD also fixes an important optimization issue in MDD, which employs an indicator function in computing $\mathcal{D}$ (viz. our Eq 13, or their Eq 30 where $h_f$ is the indicator according to their Eq 1). Since the indicator function is not differentiable, proper backpropagation is blocked.  We tried removing it in Eq 13 (i.e., just using the soft label $p$), but "observed negative infinity even after finely tuning the step size" (line 160).  This reinforces the significance of our bi-level reformulation.
>
> **Notation**:
>
> 1\. Yes, it is a typo.  Thank you for pointing this out.  Indeed, we define $P := \phi$ # $S_x$ and $\tilde{P}:=\phi$ # $\tilde{S}_x$.
>
> 2\. Yes, $Q := \phi$ # $T_x$ and $\tilde{Q}:=\phi$ # $\tilde{T}_x$.
>
> 3\. In line 83, we understand that only $\tilde{T}_x$ is available at training.  In this particular context,  transductive learning is contrasted with inductive learning whose task is to predict well on the target domain $T$.  So we wrote $\tilde{T}$ as the transductive counterpart of $T$.  To be more rigorous, we will revise the manuscript into "concerned with the prediction on the empirical distribution $\tilde{T}$, whose feature component $\tilde{T}_x$ is available at training time."
>
> We hope the motivation and notation have now been sufficiently clarified.  Please let us know if anything is still unclear.  Once more, we greatly appreciate your review and efforts.

---

> > ### Comment · Reviewer_dRho · 2021-08-21
> > **Reply to authors**
> >
> > Thanks for the detailed reply. The clarifications were helpful and I have accordingly increased my score from 4 to 6. I would request the authors to add these details in the paper for an easier reading.

---

### Official Review · Reviewer_iSrE · 2021-07-16

**Rating:** 6
**Confidence:** 4

**Summary:**

The paper proposes warping probability discrepancy measures towards the end tasks. To achieve this, the authors leveraged the pseudo-labels from the optimal predictor with a new bi-level optimization-based approach.

**Limitations And Societal Impact:**

Yes

**Main Review:**

### Originality

The proposed framework is designed upon maximum mean discrepancy (MMD) and contrastive domain discrepancy (CDD). However, the modification is technically sound and novel.

### Quality

In Table 2, ASAN is not the state-of-the-art method. [ref1] achieves 71.8% on Office-Home dataset.

In Table 3, ASAN is not state-of-the-art method. It is encouraged to include recent state-of-the-art methods, such as [ref2] and [ref3]. [ref2] achieves 90.3% and [ref3] achieves 90.2%.


### Clarity

The paper is well written and easy to follow. The highlighted words were also helpful for understanding equations.

### Significance

The experimental results of the proposed method are not significant. For example, [ref1] outperforms the proposed method by 1.0% on Office-Home and [ref2] outperforms the proposed method by 0.9%.

[ref1] Do We Really Need to Access the Source Data? Source Hypothesis Transfer for Unsupervised Domain Adaptation, ICML 2020.

[ref2] Unsupervised Domain Adaptation via Structured Prediction Based Selective Pseudo-Labeling, AAAI 2020.

[ref3] Discriminative Feature Alignment: Improving Transferability of Unsupervised Domain Adaptation by Gaussian-guided Latent Alignment, Pattern Recognition 2021.


**Time Spent Reviewing:**

6

---

> ### Author Response · Authors · 2021-08-09
> **Empirical comparison and intellectual merit**
>
> Thanks for your constructive review.
> We summarize the performance (accuracy) of [Ref1] to [Ref3] on all the datasets as follows, in comparison with i-CDD:
>
> | Method      | Office-31 | Office-Home  |  ImageCLEF  |
> |---------------|:-----------:|:------------:|:-----------:|
> | Ref1      | 88.6       | 71.8   |  88.5 |
> | Ref2   | 89.6        | 71.0      | 90.3|
> | Ref3   | 88.8        | 69.2      | 90.2|
> | i-CDD   | 90.9 | 70.8 | 89.4 |
>
> We conducted the experiment for Ref1 on ImageCLEF, and the results for each domain are as follows:
>
> |       I -> P      | |   P -> I    ||     I -> C     ||    C -> I     ||     C -> P      ||       P -> C      |
> |:------------------:||:------------------:||:------------------:||:------------------:||:------------------:||:------------------:|
> | $77.4 \pm 0.5$ ||  $92.2 \pm 0.6$  ||   $96.1 \pm 0.2$ ||   $91.7 \pm 0.4$ ||     $77.6 \pm 0.6$ ||     $95.8 \pm 0.4$ |
>
> The rest of the results are quoted from the original paper, after checking manually on the data and their code.
>
> So our i-CDD outperforms all these methods on Office-31.
> In addition, [Ref1] is inferior to i-CDD on ImageCLEF, and
> [Ref3] is inferior on Office-Home.
> [Ref2] is almost the same as i-CDD on Office-Home.
> In addition, [Ref2] requires solving a large generalized eigenvalue systems in their Eq 7.
> According to their Section "Computational Complexity",
> the cost is $O(d_1 (d_1^2 + n^2))$ for $n$ images in the source and target domains combined,
> and $d_1$ can be as large as $1024$.
> So it is highly intensive in computation for large $n$.
> Although stochastic PCA could be applied, its impact on the performance remains unclear.
>
> To conclude, our i-CDD performs very competitively overall.
> We humbly find it overly demanding to require a method outperform  state of the art on **all** datasets.
> Comparison with SOTA is only one evaluation criterion,
> and it is equally important to assess the intellectual merit in the following sense.
>
> 1\. Many recent works in UDA rely heavily on engineered tricks and heuristics customized for the benchmarks (we mentioned a few in lines 155-160 and 197-201).
> Typically they are glossed over in the paper or even not mentioned,
> and can be unearthed only by perusing the code.
> Although the empirical performance is important, we find it crucial to develop a **principled** approach that works generally well without any cherry-picked heuristics.
> Our implementation (URL provided in the supplementary) faithfully follows the description of the paper without any hidden heuristic, and the optimization simply uses the off-the-shelf Adam with minor tuning of the step-size.
>
>
> 2\. A paper's intellectual merit can be best appraised in the context of literature with similar motivations.  Although a structured SVM can be compared against a hierarchical nonparametric Bayesian approach in terms of prediction accuracy, the substantial difference in their methodology makes such a comparison less meaningful than against other frequentist methods.
> Our method is motivated from a task-driven probability discrepancy measure,
> and when applied to UDA, the ideal competitors include the class of methods based on feature adaptation.
> In contrast, [Ref1] does not consider alignment in the feature space.
> Furthermore, it learns a feature extractor for **each** individual domain to compensate for the inaccessibility to the source domain data after pre-training,
> leaving the comparison on a different ground against [Ref2, Ref3] and ours.
> [Ref2] proposed a very ad-hoc procedure instead of a principled  optimization,
> preventing the backbone features from being fine-tuned via backpropagation.

---

> > ### Comment · Reviewer_iSrE · 2021-08-26
> > **Thanks for the detailed response**
> >
> > I appreciate the authors' efforts in clarifying the merits of the proposed method over the other recent works. I have accordingly updated my score to 6 (weak acc).

---

### Official Review · Reviewer_Fjy7 · 2021-07-22

**Rating:** 6
**Confidence:** 4

**Summary:**

This paper introduces an algorithm for invariant representation learning for domain adaptation, which builds on the existing work by MDD in order to effectively calculate the $D_{\mathcal{H}\Delta \mathcal{H}}$ divergence using the source domain classifier.

**Ethics Review Area:**

["I don’t know"]

**Limitations And Societal Impact:**

No limitations are discussed, and any inclusion of them in the paper would be welcome.

**Main Review:**

The paper is clearly presented and plenty of relevant related work is cited. The paper provides a meaningful discussion on the drawbacks of MDD and addresses the problem by introducing a bi-level optimization approach. Furthermore, the paper provides training strategies to improve performance (such as taking advantage of a linear $h$, how to make use of pseudo-labels, and cache-augmented training). The empirical experiments are comprehensive and seem convincing.

One thing that would improve the paper would be a discussion on how DANN [Ganin et al., 2016] estimate the $D_{\mathcal{H}\Delta \mathcal{H}}$ divergence using adversarial training, and why it is necessary to have equation (13) when the divergence was initially minimized using adversarial training.

**Time Spent Reviewing:**

4

---

> ### Author Response · Authors · 2021-08-09
> **DANN v.s. $H \Delta H$ in Eq 13**
>
> We are grateful for your constructive reviews and thank you for recognizing our paper's intellectual merit and empirical evaluation.
>
> The divergence used in adversarial training for DANN or GAN is effectively the Jensen-Shannon divergence, viz. Eq 6 in Goodfellow et al., (2014).
> Eq 13, which was introduced by Zhang et al. (2019) in Section 4.2 of [23], tries to approximate the $\mathcal{H} \Delta \mathcal{H}$-divergence from Ben-David et al., (2010) [16].
> The Jensen-Shannon divergence is a generic measure on two distributions, while the $\mathcal{H} \Delta \mathcal{H}$-divergence is more tailored for classification problems with sharper generalization bounds as established by [16, 23].
> However, the latter raises computational issues such as non-differentiability of indicator function and exploding/vanishing gradients.  This motivated the approximation by [23] and [3], leading to Eq 13 in our paper which is the same as Eq 30 in [23].

---

### Decision · Program_Chairs · 2021-09-27

**Decision:**

Accept (Poster)

**Comment:**

This paper originally had originally some borderline negative reviews where the reviewers acknowledged a novel DA approach and  theoretical results but  noted some lack of positioning with respect to other DA methods such as DANN and some missing comparisons and justifications for the method. The authors did a very good job at answering those concerns in short and efficient replies that clarified all the major concerns from the reviewers.

The final consensus is that the paper should be accepted. Note that the authors have to take into account the comments from the reviewed and integrate their very pertinent answers to those comments in the final version of the paper.